

# Surface water floods in Switzerland: what insurance claim records tell us about the caused damages in space and time

Daniel B. Bernet[1], Volker Prasuhn[2], and Rolf Weingartner[1]

[1]Institute of Geography & Oeschger Centre for Climate Change Research & Mobiliar Lab for Natural Risks, University of Bern, Bern, Switzerland
[2]Agroscope, Research Division, Agroecology and Environment, Zurich, Switzerland

*Correspondence to:* Daniel Bernet (daniel.bernet@giub.unibe.ch)

**Abstract.** Surface water floods (SWFs) have received increasing attention in the recent years. Nevertheless, we still know relatively little about where, when and why such floods occur and cause damages, largely owed to a lack of data, but to some degree also because of terminological ambiguities. Therefore, in a preparatory step, we summarize related terms and identify the need for unequivocal terminology across disciplines and international boundaries in order to bring the science together.

Thereafter, we introduce a large (n=63'117), long (10–33 years) and representative (48 % of all Swiss buildings covered) data set of spatially explicit Swiss insurance flood claims. Based on registered flood damages to buildings, the main aims of this study are twofold: First, we introduce a methodology to differentiate damages caused by SWFs and fluvial floods based on the geographical location of each damaged object in relation to flood hazard maps and the hydrological network. Second, we analyze the data with respect to their spatial and temporal distributions aimed at quantitatively answering the fundamental

questions of how relevant SWF damages really are, as well as where and when they occur in space and time.

This study reveals that SWFs are responsible for at least 45 % of all flood damages to buildings and 23 % of the associated direct tangible losses, whereas lower losses per claim are responsible for the lower loss share. The Swiss lowlands are affected more heavily by SWFs than the alpine regions. At the same time, the results show that the damages are not evenly distributed within each region either. By far the most SWF damages occur during summer in almost all regions. The normalized damages

of all regions show no significant upward trend of SWF damages between 1993–2013. We conclude that SWFs are in fact a highly relevant process in Switzerland that should receive similar attention like fluvial flood hazards. Moreover, as SWF damages almost always coincide with fluvial flood damages, we suggest to consider SWFs, just as fluvial floods, as integrated processes of our catchments.

## 1 Introduction

In Switzerland, there seems to be a growing awareness that just as overtopping rivers and lakes pose substantial flood risks for society, so too does flooding that takes place far away from watercourses. All across Europe, there are well-known examples of such inland flood events. In 1988, for instance, a devastating flood occurred in Nîmes, France (e.g., Davy, 1990; Andrieu et al., 2004). In 2007, Hull UK was affected by flooding (e.g., Pitt, 2008; Coulthard and Frostick, 2010). One year later Dortmund, Germany, experienced wide-spread flooding (e.g., Grünewald, 2009). In 2011, the Danish capital Copenhagen was affected



heavily by flooding (e.g., Haghighatafshar et al., 2014). The Swiss canton of Schaffhausen was affected severely in 2013 (e.g., Scherrer et al., 2013) and the Dutch capital Amsterdam experienced substantial flooding in 2014 (e.g., Gaitan et al., 2016). These events in Europe share a common thread, which stems from their origin as inland floods, triggered by heavy precipitation, but mostly unrelated to watercourses.

As the definition of such floods is not straightforward, we adopt the term of surface water floods (SWFs) for now, use it for non-fluvial floods in general and discuss the terminology in Sect. 2 in detail. Inherently, SWFs are not constrained to areas close to watercourses, but can occur practically anywhere in the landscape (Kron, 2009). Consequently, such floods are difficult to document, study and forecast (e.g., Pitt, 2008; Steinbrich et al., 2016) and related data are scarce (e.g., Hankin et al., 2008; Douglas et al., 2010; Blanc et al., 2012; Grahn and Nyberg, 2017). Spekkers et al. (2014) mention the lack of data as well as the

impact on small spatial scales as possible explanations why relatively little scientific research has been dedicated to such SWF in comparison to fluvial floods. In contrast, gray literature covers the topic of SWFs rather extensively, which is reflected by the availability of many guidelines and manuals discussing how to prepare for and manage such floods, for instance for single objects in Switzerland (Egli, 2007), or on communal or regional levels in Germany (e.g., Castro et al., 2008; DWA, 2013; LUBW, 2016) or France (e.g., CEPRI, 2014). This might exemplify that the scientific flood risk community is indeed quite

oblivious of resourceful gray literature (Uhlemann et al., 2013). In any case, it indicates that the topic is a concern for the people, the responsible authorities and other stakeholders. In order to reduce the risk, it is suggested to focus on the physical protection of exposed objects (e.g., Kron, 2009; DWA, 2013). Although this strategy is certainly pointing into the right direction, we have to be conscious about the basis on which current and future decisions concerning SWFs are taken. Undoubtedly, the lack of quantitative data and studies hampers our process understanding (Grahn and Nyberg, 2017). Therefore, the underlying crucial

question is "how can we reduce losses from natural hazards when we do not know (...) when and where they occur?" (Gall et al., 2009).

Owing to vast river discharge time series, fluvial floods can be well predicted along gauged rivers (Steinbrich et al., 2016). As there are no such data concerning SWFs (Steinbrich et al., 2016), we must exploit other data sources in order to quantify the relevance of this flood type in space and time. Possible data sources include, but are not limited to, insurance claim records

(e.g., Spekkers et al., 2013; Zhou et al., 2013; Moncoulon et al., 2014; Bernet et al., 2016; Grahn and Nyberg, 2017), disaster data bases (e.g., Gall et al., 2009; Kron et al., 2012), press reports (e.g., Hilker et al., 2009) and interviews with or reports from affected people (e.g., Thieken et al., 2007; Evrard et al., 2007; Gaitan et al., 2016). All data sources are probably subjected to a varying degree of a so-called "threshold bias", which refers to the bias introduced due to varying damage inclusion criteria (Gall et al., 2009). Disaster data bases only list events that exceeded predefined loss and/or fatality thresholds (Kron et al.,

2012). Similarly, damage data based on news reports are subjected to unknown thresholds, as damages are only reported if they are found to be interesting enough. As interview campaigns are more likely to be initiated after devastating flood events, such data are biased towards more extreme events, as well (Elmer et al., 2010). Insurance claim records are likely affected the least by a threshold bias, as long as the related insurance policy stays the same, insured objects are not changing greatly over time and the deductibles are low or can be accounted for.



Damage claim records of insurance companies are therefore a profitable data source. Not surprisingly, they have been the base for several studies related to SWFs (e.g., Cheng et al., 2012; Spekkers et al., 2013; Zhou et al., 2013; Moncoulon et al., 2014; Spekkers et al., 2015; Bernet et al., 2016; Grahn and Nyberg, 2017). Unfortunately, insurance claim data are generally difficult to collect, since most insurance companies do not publish or provide loss data due to confidentiality issues (Boardman, 2010; Grahn and Nyberg, 2017). Furthermore, analyses based on such data are often impaired by the data's spatial or temporal aggregations. For instance, the limited usefulness of monthly aggregated data was demonstrated by Cheng et al. (2012). On the other hand, Spekkers et al. (2014) pointed out some limitations of insurance data aggregated to administrative units, which do not have homogeneous topographical properties. Moreover, it is difficult to differentiate and verify the cause of each damage without knowing the explicit location of the damaged object. This is particularly important, as the corresponding data often cover different processes without explicit classification: For instance, Grahn and Nyberg (2017) had to exclude all damage records with dates that coincided with dates of known fluvial flood events to obtain a subset of SWF-related claims. Spekkers et al. (2013) chose a more elaborate methodology by applying a statistical filter based on the assumption that rainfall-related damages are clustered around wet days, while other causes are occurring on any day throughout the year. Finally, even though many or even all buildings are insured against floods in several countries (e.g. in Sweden (Grahn and Nyberg, 2017) or in the Netherlands (Spekkers et al., 2014)), usually only a subset of all objects are covered by the obtained data records. This is owed to the fact that the objects are usually insured by many different companies, each having a different (unknown) market share. In addition, these shares are generally not constant over time either, but may fluctuate heavily over time and space, as exemplified by Spekkers et al. (2014). These spatial and temporal changes need to be taken into account, which is often not trivial.

Luckily, most of these limitations are not applicable for damage claim records of the Swiss public insurance companies for buildings (PICB). In Switzerland, PICB are present in 19 out of the 26 cantons, whereas each company insures (almost) all buildings within the respective canton due to their monopoly position and because the insurance is generally mandatory for all house owners. Beside other natural hazards, the insurance covers damages caused by floods, which includes both fluvial floods as well as SWFs. Data records of PICB are, therefore, exceptionally interesting for analyzing floods in general, and SWFs in particular. Most PICB have shown a general interest about research on this topic and, thus, were willing to provide flood claim records including the address of each damaged object.

Based on these data, the first aim of this study is to provide a methodology, with which each claim can be classified as being caused by SWFs or fluvial floods, respectively. Second, based on the classified claim records, we aim at answering the fundamental question of how relevant damages caused by SWFs are, as well as where and when these damages occur in space and time. The underlying data set stems from 13 PICB and covers 48 % of all buildings in Switzerland. Thus, the data set is representative for most of Switzerland, except for southern Switzerland (i.e. Western Inner Alps and Southern Alps). The analyzed data records all end in 2013 and extent back to at least 2004, but even up to 1981 depending on the PICB. As the PICB, safe a few exceptions, insure only property and not its contents, this study considers only damages to buildings. Moreover, this study is limited to direct tangible flood losses, although we acknowledge that these only constitute part of the total flood losses.

We have identified a lack of a common terminology concerning SWFs. Therefore, we dedicate the following Sect. 2 to a short overview of terms that are currently being used to address flood types that could be categorized as SWFs, as mentioned



before. In Sect. 3 we describe the data in detail and introduce the methodology to differentiate SWF damages from fluvial flood damages. Thereafter, in Sect. 4, we present general characteristics of the number of claims as well as associated loss caused by SWFs in comparison to fluvial floods. Furthermore, we present the spatial and temporal characteristics of damages caused by SWFs in Switzerland during the last decades and discuss the results in Sect. 5. Finally, by providing concluding remarks, we conclude the study (Sect. 6).

## 2 Terminology

As precipitation reaches the land surface, different runoff generation mechanisms determine whether surface runoff is generated (e.g., Fiener et al., 2013). The water may then take several routes towards the stream channels (Ward and Robinson, 2000), as depicted in Fig. 1. The flow path along the land surface is sometimes ambiguously referred to as "surface runoff", but is better defined by the widely-used term "overland flow" (Ward and Robinson, 2000). However, in the literature, this distinction is inconsistently made, whereas either of the terms or even both are used. We adopt the term overland flow and, thereby, mean the transport of water downhill at the land surface as thin sheet flow or anastomosing braids of rivulets and trickles until the water reaches or is concentrated into recognizable streams (Chow et al., 1988; Ward and Robinson, 2000; Brutsaert, 2005). The propagation and accumulation (i.e. ponding) of overland flow can be considered as a flood, which in the glossary of Field et al. (2012) is defined as "the overflowing of the normal confines of a stream or other body of water, or the accumulation of water over areas that are not normally submerged". However, we have to keep in mind that the term "flood" is sometimes implicitly used in the hydrological sense, but sometimes also in the context of "damaging floods" (Barredo, 2009). In the former case, any inundation of land is considered, while in the latter case, the flood necessarily interacts with the societal system causing adverse effects (Barredo, 2009). Obviously, damage data are inherently only reflecting damaging floods. Consequently, we have to bear in mind that by using damage data, we can only draw direct conclusions about damaging floods, and not hydrological floods in general.

Kron et al. (2012) noted the importance of consistently using well-defined terms for addressing the perils listed in disaster data bases. This is not just true for data, but for scientific research in general. For instance, Boardman (2010) has identified the lack of terminology as a possible reason for the accounts of specific flooding (i.e. muddy flooding) in some countries and the seeming non-existence in other areas. As this might also partly explain why SWFs have received attention in some countries but remain unrecognized in others, we have identified the need for a short terminological elaboration, following hereafter.

SWFs are caused by intense rainfall that, due to whatever reason, cannot be drained by natural or artificial drainage systems and, thus, ponds in local depressions or propagates along the surface as overland flow (Pitt, 2008; Hurford et al., 2012), before it possibly, but not necessarily, reaches or is concentrated into regular watercourses (Fig. 1). The term is often used synonymously with "pluvial floods", although according to Falconer et al. (2009), SWFs have a broader meaning. Namely, the term does not only include pluvial flooding, but also flooding from the sewer, small open or culverted watercourses as well as flooding from groundwater springs (Hankin et al., 2008; Falconer et al., 2009). Therefore, SWFs can be thought of as the most general form of rainfall-related (non-fluvial) floods. In this study, we analyze damages caused by SWFs, as defined above.





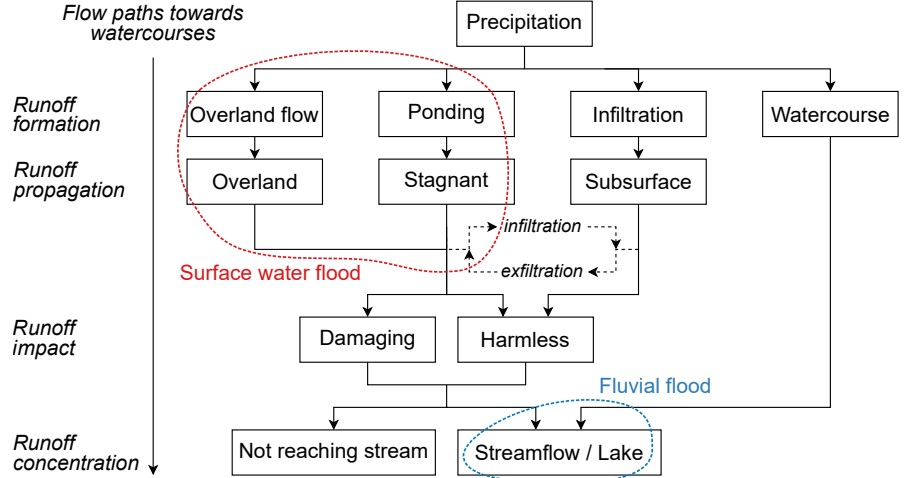

**Figure 1.** As precipitation reaches the land surface, it may directly fall into a stream or take different routes towards it, governed by the relevant runoff generation mechanism. Thereby, overland flow and ponding constitute a SWF that has not yet or will never reach a watercourse. In contrast, a possible fluvial flood originates from the body of water itself, when the normal confinements are overflowed. Moreover, we note that all floods may or may not cause damages. Understandably, studies based on damage data analyze a subset of all floods, i.e. only those for which some sort of damage has been registered.

The term "flash flood" is sometimes used in relation to floods that are unrelated to watercourses (e.g., Castro et al., 2008; Kron, 2009; Kron et al., 2012; DWA, 2013; Steinbrich et al., 2016). However, the definition of flash floods is generally quite ambiguous (van Campenhout et al., 2015). Mostly, it refers to fluvial floods triggered by short, intense and local storm events (e.g., Merz and Blöschl, 2003; Gaume et al., 2009; Falconer et al., 2009; Ruiz-Villanueva et al., 2012), but the term flash flood

may include other causes as well (Castro et al., 2008; Priest et al., 2011; Gourley et al., 2013).

According to Andrieu et al. (2004) and Douguédroit (2008), the main difference between flash floods and "urban floods" is that in the former case the flood originates from watercourses, while in the latter case overland flow is generated within the urbanized area itself. Urban floods are also referred to as an "intra-urban floods" (Evans et al., 2004; Hankin et al., 2008) and flash floods affecting urban areas are sometimes called "urban flash floods" (e.g., Hankin et al., 2008; DWA, 2013; Zhou et al.,

2013) or are even used synonymously to urban floods (Kron et al., 2012), owing to a broad definition of flash floods in the first place.

Another non-fluvial flood type is "muddy flooding". According to Boardman (2010), such floods are formed by muddy runoff from agricultural fields that damage adjacent properties downslope. The term is well-established (Ledermann et al., 2010).

Overall, many of the terms used to address different flood types are either used ambiguously in the literature or are not well-defined. Flooding is a complex interlinked system, affecting many aspects of the physical, economic and social environments acting at different spatial and temporal scales (Evans et al., 2004; Barredo, 2009). As such, flooding involves a wide range of



interconnected hydraulic subsystems and processes (Evans et al., 2004). Therefore, it is understandable that the classification of such a complex process like flooding is not simple, particularly in practice. It seem all the more important that the used terms and the corresponding definitions are well documented. This might reveal terminological ambiguities and, ultimately, make meaningful comparisons possible (Gall et al., 2009).

## 3    Materials and methods

The compiled data set is based on flood damage claim records from 14 different PICB. In addition, we obtained similar records from the cooperative insurance company Swiss Mobiliar, which are not part of this study's data analyses. Yet, they are used to set up the claims' classification scheme alongside the data from the PICB (c.f. Sect. 3.1). As mentioned before, each PICB holds a monopoly position and, thus, insures virtually every single building within the respective canton against various natural hazards including flooding. Thereby, damages caused by water entering the building envelope at the surface are insured, while damages associated with direct intrusion of groundwater or backwater from the sewer, as well as flooding from dams or other artificial water structures are generally excluded. In consequence, water-related damages covered by PICB are either caused by SWFs or fluvial floods, whereas the insurance companies themselves do not differentiate the two processes (Imhof, 2011). Therefore, similar to other studies (e.g., Spekkers et al., 2013, 2015; Grahn and Nyberg, 2017), the data have to be classified first. However, in contrast to the aforementioned studies, the claim records were provided in a spatially explicit way, enabling a classification based on each claim's geographical context.

Following the data processing procedure depicted in Fig. 2, we first describe the compiled data set as well as the harmonization and geocoding thereof (Sect. 3.1). Following, we introduce the methodology to differentiate claims associated with SWFs and fluvial floods (Sect. 3.2) and, thereafter, we discuss the necessary normalizations of the data (Sect. 3.3). Note that the classification scheme is described as generally as possible, to make its application to other contexts and countries as straightforward as possible. However, it could not be prevented that the classification scheme is adapted to some national characteristics, in particular concerning the properties of the considered Swiss flood maps. The specific input data for each data processing step listed in Fig. 2 are described in detail in Table 1.

### 3.1    Data

Figure 3 gives an overview of the compiled data set and illustrates all 19 cantons with a PICB, while the 14 PICB that provided data are highlighted additionally. As the cantons' borders have mostly administrative meaning, we adopted the natural landscape units from Grosjean (1975), while constraining the borders to hydrological catchment boundaries. In this study, the data are analyzed with respect to these regions (Fig. 3). Overall, 43–100 % of the buildings are covered by our data set, with the exception of the Western Inner Alps (0 %) and the Southern Alps (6 %). The low values of the latter two regions are owed to the fact that practically no buildings are insured by a PICB within these areas. Consequently, these areas are excluded from this study's analyses, even though some claims provided by the Swiss Mobiliar covered this region. The provided data provided by the Swiss Mobiliar contain flood damage claim records of content and, additionally, of property in cantons with no PICB.





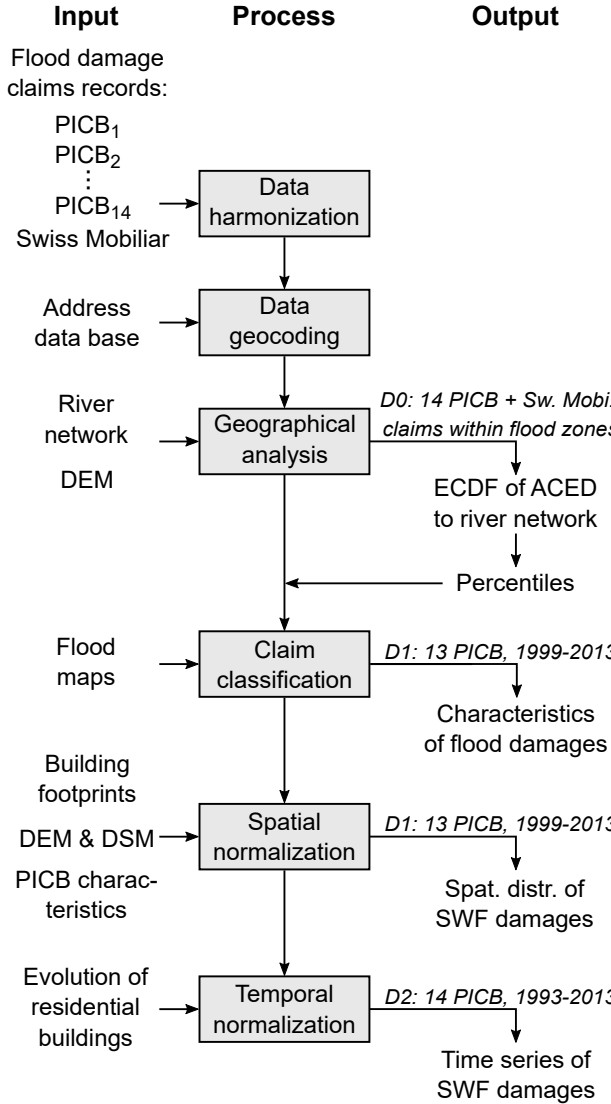

**Figure 2.** Illustration of the main data processing steps (boxes) as well as the required input data, which are further specified in Table 1. *D0*, *D1* and *D2* refer to the data (sub-) sets, which were used to produce the output, illustrated by this study's tables and figures. Note that *D0* constitutes the complete data set including data from 14 PICB in addition to data from the Swiss Mobiliar, whereas *D1* and *D2* consist of PICB data only, limited to the indicated periods (c.f. Table 2). The empirical cumulative distribution function (ECDF), as well as the altitude constrained Euclidean distance (ACED) between each claim and the next river are abbreviated (c.f. Sect. 3.2).

These records have quite similar characteristics as the data provided by the PICB, but are not limited to certain cantons and, thus, extend over the whole of Switzerland. However, as pointed out in the introduction, records of insurance companies with certain (unknown) market shares are much more challenging to interpret. Nevertheless, the data are useful for setting up the





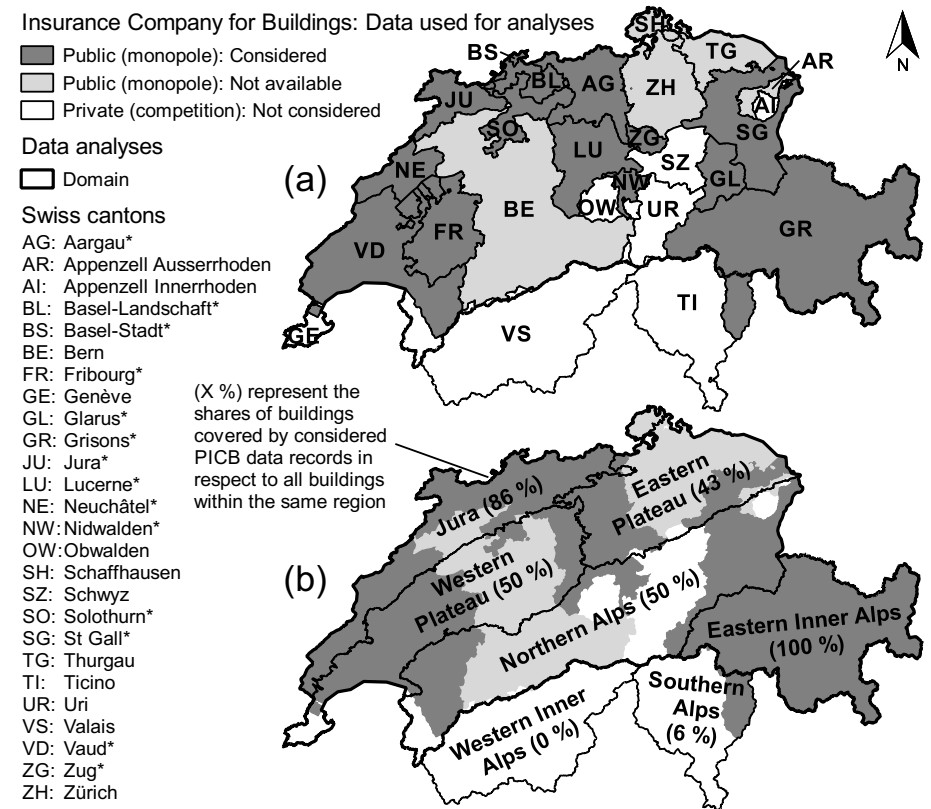

**Figure 3.** Overview of the compiled data set *D1* (c.f. Table 2). **(a)** Cantons with and without a PICB, as well as an indication which PICB provided data. The latter are additionally marked with an asterisk (*) in the legend. **(b)** Natural landscape units based on Grosjean (1975), which are used to analyze the data on a regional scale. As almost no buildings are insured by PICB within the Western Inner Alps as well as the Southern Alps, these two regions are excluded from the data analyses, as indicated by the domain.

classification scheme, because every additional claim generally increases the method's robustness (c.f. Sect. 3.2). The data from the Swiss Mobiliar are included in the data set *D0*, which is used for the classification scheme (c.f. Table 2).

The minimal information of each flood damage claim includes the damage date, the location of the damage (address or coordinates) as well as the associated direct tangible loss to the respective building. As the claim data stem from 15 different
5  data sources (14 PICB and data from the Swiss Mobiliar), the provided raw data are heterogeneous and need to be harmonized first, as indicated in Fig. 2 (c.f. Bernet et al., 2016, for details). During this procedure the data were quality checked, obvious errors were corrected, if possible, or removed otherwise. Moreover, the loss values were corrected for inflation as of 2013 using index values of the corresponding PICB, in case the source data had not been indexed already.

During the next step, each damage claim is geocoded (Fig. 2). The coordinates of each damaged building could be obtained
10  by matching the corresponding address with a geocoded register of all Swiss postal addresses (c.f. Table 1). Notably, only the claims with an unique match were analyzed later. As the data quality of the addresses varies among the different PICB, the





amount of claims that could be localized at the building level varies as well (Table 2). Nevertheless, most of all PICB claims (79 %) could be localized. A summary of the compiled (sub-) data set is given in Table 2.

## 3.2 Classification

The basic idea behind the classification scheme is simple: In case a building (and/or its content) has been damaged by flooding
and was located far away from any watercourse, it is very likely that the damage was caused by a SWF. The opposite is not necessarily true: Overland flow is propagating over the land surface towards the watercourses and might cause damages along the flow path until it reaches the next watercourse (Fig. 1). Thus, for damaged objects close to a watercourse it is difficult to deduce the responsible flood type without studying each case in detail. Given the size of the data set, detailed manual classification is not practical, in addition to the fact that the data generally do not contain additional information about the
responsible damage causes.

In order to classify the claims pragmatically, we exploit the damages' known locations as follows: We assume that the dominant damage process in known fluvial flood zones are fluvial floods and, thus, damaged objects located within such zones were likely affected by this process. As these damages are inherently clustered around watercourses, we make use of this characteristic by assessing the distance between these damages and the next river. We then classify the claims outside of
known flood zones based on how their own distance to the next river relates to the typical distances obtained from fluvial flood damages. Thereby, the question is how these distance should be measured and how a representative cut-off distance can be determined.

We tested different distance measures, whereas the Euclidean distance performed well for instance, but neglected topography altogether. For instance, a building on a ridge can be associated with a short Euclidean distance to the next river, in spite of
being safe from river flooding due to the building's elevated location. We therefore chose the following approach to address this issue, while at the same time making use of the Euclidean distance's simplicity: Before calculating the Euclidean distance to the next river, we first masked the river network with areas lower than the respective object's altitude using a digital elevation model (DEM, Table 1). Only then, the Euclidean distance to the masked river network is assessed. The obtained quantity is hereafter referred to as "altitude constrained Euclidean distance" (ACED).
Typical distances for all fluvial flood damages can then easily be obtained by analyzing the ACEDs of all claims located within known flood zones. For that matter, we selected all claims within such flood zones and compiled the empirical cumulative distribution function (ECDF) of the ACEDs. Based on the large data set, we can be confident that the claims located farther away from the closest river than the 99th percentile of the respective ECDF were caused by SWFs. Considering that fluvial floods become generally more probable the closer we get to the rivers, we chose evenly spaced percentiles, i.e. the 25th,
the 50th, 75th and 99th percentile, respectively. To take regional geographical characteristics into account, the percentiles are calculated for each region separately (Table 3).

Inherently, flood claims also include damages caused by overflowing lakes, which could not be distinguished easily from fluvial floods. Consequently, damages related to lakes will be associated with a certain distance to the next river, even though the corresponding river was not the cause of the damage. As this rather tends to shift the ECDF to the right, in addition to the



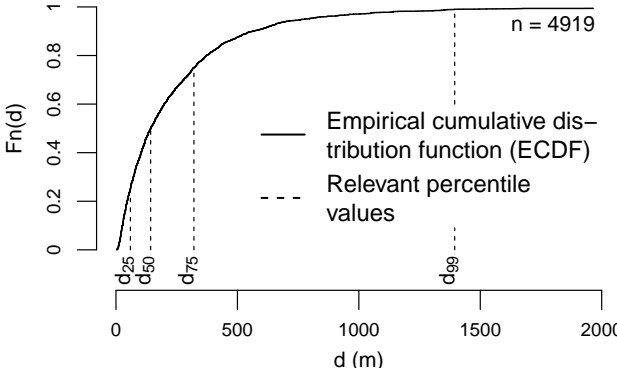

**Figure 4.** ECDF of all ACEDs of the claims within flood zones in the region Jura. Such ECDFs were compiled separately for all five analyzed regions in Switzerland (c.f. Fig. 3). The corresponding percentiles are used for the classification of the claims (Fig. 5), and are listed in Table 3.

low number of such cases, it is safe to assume that this influence is negligible. Moreover, the hazard of overflowing lakes is consistently considered in the fluvial flood maps. Consequently, the damages caused by overflowing lakes are located within mapped flood zones and are, therefore, directly and correctly classified as fluvial floods (c.f. Fig. 5).

Using the precompiled percentiles (Table 3) as well as fluvial flood maps (Table 1) as input, the damages can then be
classified by means of the classification scheme presented in Fig. 5. Five different classes are differentiated ranging from *most likely surface water flood (A)* to *most likely fluvial flood (E)* (c.f. Fig. 5). The qualitative confidence levels reflect that in general it is becoming gradually more unlikely that an object is affected by fluvial floods the farther away an object is located from a river.

As outlined in Fig. 5, we make use of two particular fluvial flood maps, i.e. the "official" Swiss flood hazard maps (Zim-
mermann et al., 2005; de Moel et al., 2009) and an ancillary map available for the whole of Switzerland called Aquaprotect (Table 1). As for the Swiss flood hazard maps, the Swiss Mobiliar collected all available maps from each canton and, in agreement with the responsible authorities, provided the data as per December 2016. The data contain the perimeters for which fluvial flood hazards have been mapped in detail. Within these perimeters, the fluvial flood hazards are indicated using four different danger levels (de Moel et al., 2009), whereas we define the flood hazard zone as the combined area associated with
low, medium and elevated danger, excluding residual danger (c.f. Zimmermann et al., 2005). As indicated by de Moel et al. (2009), the flood hazard maps are available for almost the entire Swiss territory. In fact, 88 % of all claims are covered by the flood hazard maps as of 2016, i.e. they are located within the hazard maps' perimeters. The number has increased rapidly over the recent years. Nevertheless, there are still cantons where more than 60 % of the claims are located outside of the perimeters. Thus, to increase the coverage, we used the aforementioned map called Aquaprotect (Table 1). It contains coarse fluvial flood
extension maps compiled for return periods of 50, 100, 250 and 500 years. We chose the map representing a return period of 250 years, as it best matches the return period of up to 300 years considered by the flood hazard maps. As indicated in Fig. 6,





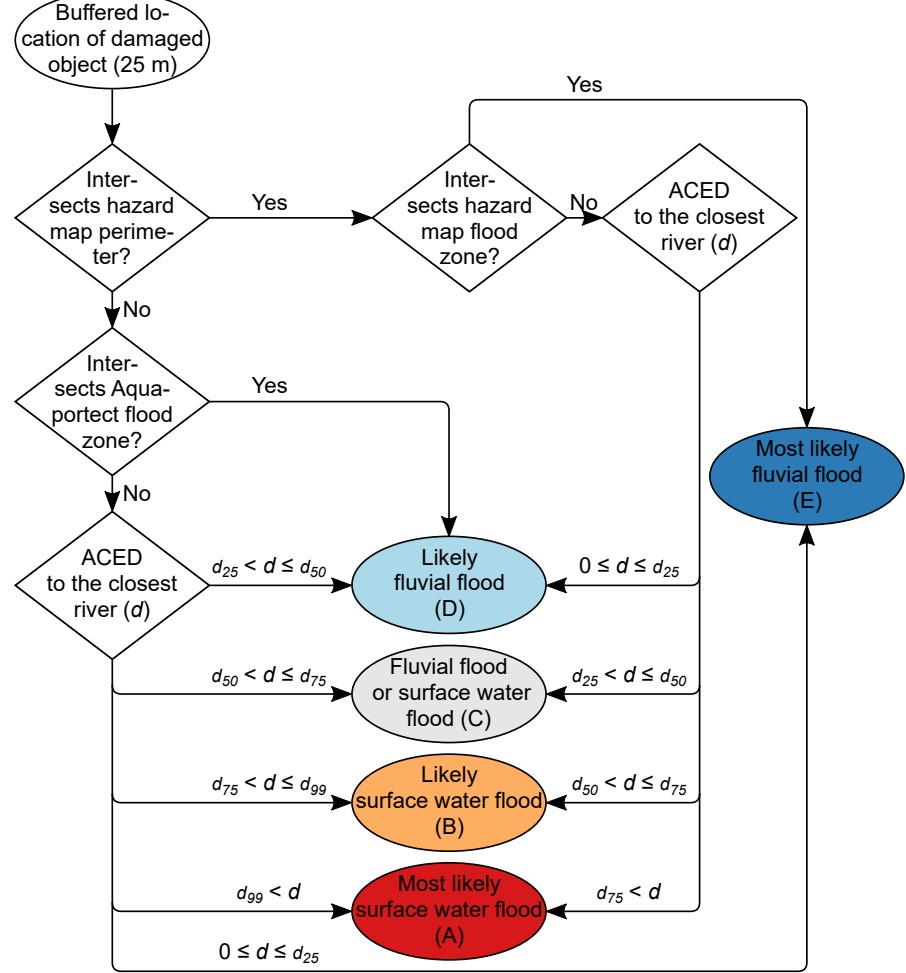

**Figure 5.** Classification scheme applied to all localized damage claims. As indicated, each claim's point location is buffered by 25 m, corresponding to an average building width. This accounts for the fact that in reality the buildings have a certain spatial extent. The claims are classified as *most likely fluvial flood (E)* in case their buffered location intersect the hazard map flood zone or as *likely fluvial flood (D)*, if it intersects the ancillary flood map Aquaprotect, respectively. The different qualitative confidence levels reflect the level of detail of the two different flood maps (c.f. Table 1). In all other cases, the specific ACED ($d$) of each claim is compared to the typical ACEDs of fluvial flood damages ($d_{25}$, $d_{50}$, $d_{75}$ and $d_{99}$, c.f. Table 3). The classification scheme is further illustrated by Fig. 6.

Aquaprotect is only used for the territory not covered by the flood hazard maps. Namely, the hazard map perimeters have been extracted from the Aquaprotect layer using common GIS tools.

It should be noted that the areas not covered by flood zones, i.e. the hazard-free zones, have similar implications for the two different sources. The smallest rivers were not considered by Aquaprotect, but there is no objective way of knowing where the cut-off was set. This also holds true for the flood hazard maps, as the study of a few examples revealed. Moreover, the





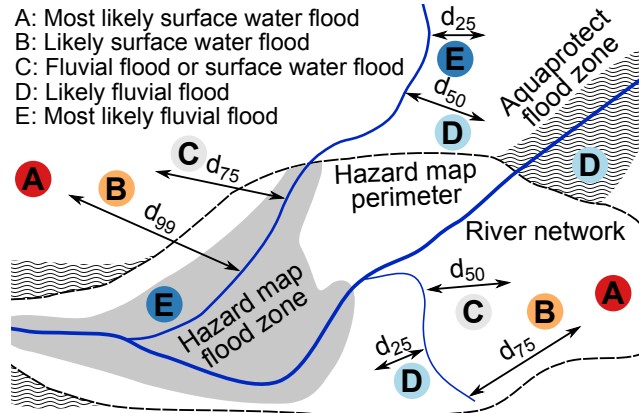

**Figure 6.** Schematic visualization of the classification scheme. Note that each of the shown classified damage claims corresponds to one of the 11 unique paths of the classification scheme depicted in Fig. 5.

flood hazard maps are produced independently by the regional governments (de Moel et al., 2009), i.e. cantons. Consequently, the applied methods vary between the different cantons and, thus, general statements cannot be made. Nevertheless, the level of detail of the Swiss flood hazard maps far exceeds the one of Aquaprotect. This is taken into account by applying lower percentile levels for claims located within the flood hazard perimeters, as shown in Fig. 5 and 6.

## 3.3 Normalizations

Reported increasing trends of flood losses (e.g., Kron et al., 2012; Grahn and Nyberg, 2017) might be misleading. In fact, there is evidence that increasing flood losses are mainly owed to socio-economic development rather than trends in the flood processes itself (Barredo, 2009). Increasing losses caused by natural hazards such as flooding can, thus, mostly be attributed to increasing population and expansion into hazardous areas (e.g., Cutter and Emrich, 2005; Barredo, 2009; Bouwer, 2011; Kundzewicz et al., 2014), as well as increasing property values and diminishing awareness about such hazards (Kundzewicz et al., 2014) and, additionally, better documentation of damages in the more recent past (Gall et al., 2009). Consequently, the loss data need to be normalized with regard to such effects, if the natural process rather than the product with the socio-economic background is of interest. The most fundamental normalization is to adjust past losses to the current values (Kron et al., 2012). However, the more difficult part is to remove the influence of socio-economic development on the observed number of damages as well as the associated loss. In addition, the consideration of a change in the exposed objects' vulnerabilities is even more difficult (Bouwer, 2011).

In this study, the values are adjusted for inflation during the harmonization procedure (Sect. 3.1). Furthermore, the absolute damage data are normalized in space by relating them to the number of buildings and the sum insured as of 2013 (Appendix A1). Finally, by normalizing the data over time (Appendix A2), we obtain a time series of normalized SWF damages. Thereby, we assume that the buildings' vulnerabilities with regards to SWFs have remained constant within the last decades. To test whether





the damages have increased or decreased over time, we apply the seasonal Mann–Kendall test (Hirsch et al., 1982) and use a significance level of 0.1 for the resulting p-value.

## 4 Results

In Switzerland, there are few quantitative studies about SWFs. One example is a case study based on damage records stemming
from the PICB of the canton of Aargau (Aller and Petrascheck, 2008), undertaken in the aftermath of the devastating August 2005 flood. The study indicated that on average at least half of the flood damages to buildings were caused by overland flow (Aller and Petrascheck, 2008), i.e. by SWFs. However, the study is neither comprehensible in terms of applied methods, nor the underlying data, and covers only a small part of Switzerland. Therefore, after a rather qualitative validation of the methodology (Sect. 4.1), we quantify, characterize and compare damages caused by SWFs with damages caused by fluvial floods (Sect. 4.2).
Following, we present the spatial distribution of SWF damages (Sect. 4.3) and show how these damages evolved within the last 20 years (Sect. 4.4).

Note that damages classified as A or B, i.e. *(most) likely surface water floods*, are regarded as damages caused by SWFs, if not stated otherwise. Analogously, damages classified as D or E, i.e. *(most) likely fluvial floods*, are counted as damages caused by fluvial floods. Claims of class C, i.e. *fluvial flood or surface water flood*, are neither counted for one, nor the other flood
type, unless total values are presented.

### 4.1 Validation

There are few data sets available with which the claims' classification or normalization could be validated. A few possibilities are elaborated hereafter.

In 2016, the canton of Lucerne published an overland flow depth map stemming from hydrodynamic simulations based on
the methodology described by Kipfer et al. (2012). However, the map indicating categorized flow depth polygons is not suitable for a quantitative validation of the claims' classification. The polygons all indicate a minimal flow depth of 0.015 m and are very dense. In fact, 67 % of all building footprints of the canton of Lucerne intersect such a polygon, whereas only 6.5 % of the footprints are farther than 10 m away from the closest polygon. Consequently, neither quantitative nor a visual relationship could be found between each claim's class and the categorized flow depths.
Two of the cantons covered by our data set provide hazard indication maps concerning overland flow, i.e. the canton of Basel-Landschaft and Aargau, respectively. However, the hazard of overland flow was not assessed comprehensibly judging from the technical reports that are publicly available. In some sub-regions, the hazard was assessed by means of GIS analysis and/or based on known past events, or the hazard was not considered at all. Consequently, these maps did not allow for a direct quantitative validation either. Namely, many claims associated with overland flow were far from any overland flow hazard
zone, likely because the hazard was not assessed or no events have been registered so far. Nevertheless, the indicated hazard zones were mostly located in the vicinity of SWF claims. This might highlight that the corresponding claims were the cause





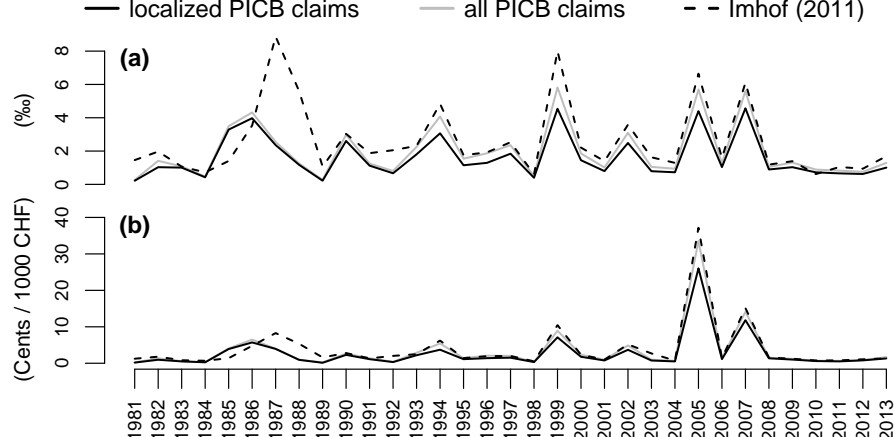

**Figure 7.** Validation of the normalized damage data. **(a)** Aggregated normalized number of claims in relation to the total number of insured buildings. As a reference, data stemming from a subset of the data presented in Imhof (2011) are shown. As the data are spatially aggregated, all the data including claims without a geocode could be shown, in addition to the localized claims. **(b)** Aggregated normalized loss in relation to the total sum insured.

for the delineation of these zones, but at the same time, it also indicates that the classification scheme produces meaningful results.

Overall, a systematic validation of the classification was not feasible, owed to the large number of claims and the lack of suitable data. Nevertheless, the classification was checked visually, drawing from the input data including flood maps, the

river network, the DEM, for example (c.f. Table 1). This manual assessment indicated that the classification scheme rendered reliable and plausible results.

Unlike for the classification, it was possible to verify the overall performance of the applied normalizations. Specifically, we could compare our normalized data set with virtually the same source data that had been normalized with the corresponding property data. The reference data root from a subset of the data shown in Imhof (2011). The reference data show aggregated

flood damages per number of insured buildings (Fig. 7a), as well as the loss per total sum insured (Fig. 7b). Thereby, the reference data consist of (almost) the complete records of the 14 corresponding PICB, whereas our data set contains less and less records, as we move back in time (c.f. Table 2). As we are looking at relative numbers, the comparison is still valid, but the different data coverages have to be kept in mind. In fact, Fig. 7 highlights that before 1989, the data sets are badly matching, but have very similar patterns thereafter. Together with the fact that after 1993 all regions are satisfactorily represented, these

are the reasons why we have limited the time series of SWF damages to the period from 1993–2013 (c.f. Table 2 and Fig. 14).

The clear bias of the localized claims in comparison to the reference data can mainly be attributed to the 21 % of the claims that could not be localized, i.e. the curves aline much better, when also considering the claims without a precise geocode (Fig. 7). However, a small bias persists, to a larger degree for the number of claims and to a smaller degree for the loss values, respectively. The remaining deviations are probably due to the coverage that becomes increasingly different in earlier years as





well as the applied normalization procedure using auxiliary data. Notably, given the simple applied methods, the normalization works exceptionally well.

## 4.2 Relevance of surface water flood damages

Figure 8 reveals that SWFs were responsible for 45 % of all localized flood damage claims between 1999–2013 based on the data set *D2* that covers 48 % of all Swiss buildings (c.f. Table 2). In terms of loss, however, SWFs only account for 23 % of the total loss. The regional loss shares vary only slightly, i.e. between 15 and 25 %, except in the Western Plateau, where SWFs account for 51 % of the total loss. In the same region, SWFs caused two-thirds of all damages. In the Jura, roughly half of all claims could be associated with SWFs. The share is lower in the Eastern Inner Alps and the Eastern Plateau with 43 % and 39 %, respectively. In the Northern Alps, SWFs are only responsible for 24 % of the flood claims.

The distribution of loss per claim explains why almost half of all claims are only responsible for roughly one-quarter of the total loss. As shown in Fig. 9, the mean loss per SWF claim is considerably lower than the mean loss per claim related to fluvial floods. This is most pronounced when comparing claims of class A (*most likely surface water floods*) with class E (*most likely fluvial floods*): For class A, 95 % of all claims are less or equal to CHF 32'349, while for class E, the 95 % percentile is CHF 120'330. Although there is a significant difference, the medians are relatively low for claims of class A and E with values of CHF 3'113 and CHF 5'554, respectively. Thus, the majority of the claims of all classes are associated with a rather low amount of loss, while the minority of the claims report extreme losses. However, by far the highest losses are associated with claims of class E (Fig. 9). Grahn and Nyberg (2017) have found similarly skewed distributions caused by pluvial floods in Sweden.

So far, an unanswered question has been, how the number of damages and associated losses are distributed in relation to the size of the corresponding event. Supposedly, frequent damages associated with low loss values might add up to a substantial sum in the end, as suggested by Kron (2009), for instance. For that matter, we have stratified the data according to the total number of claims per day using five categories ranging from *single* (1–5 claims per day) to *vast* (>501 claims per day). We defined an event as a day with at least one claim of any class (A–E), which amounts to a total of 1'490 events in the period of 1999–2013. Obviously, this is a pragmatic definition of an event, but it serves the purpose of a first simple analysis.

The stratified number of claims (Fig. 10a, *Total*) confirms that smaller events are more frequent than larger events, i.e. 1'100 events of the smallest category (*single*) are opposing 11 events of the largest category (*vast*). Interestingly, days with *single* and *few* claims only account for a small share of SWF and fluvial flood claims, although for SWFs the shares are larger. Strikingly, 11 events within the last 15 years with more than 500 claims each, account for almost half of the claims caused by fluvial floods, but only to one-quarter of the claims associated with SWFs. In contrast, the same 11 events accounted for 45 % of the loss caused by SWFs, and even 76 % of the loss caused by fluvial loss, respectively (Fig. 10b). Based on this analysis, we can infer some important characteristics about the damages caused by SWFs:

- SWF damages occur more frequently during small events, whereas the majority of fluvial flood damages are caused during large events.





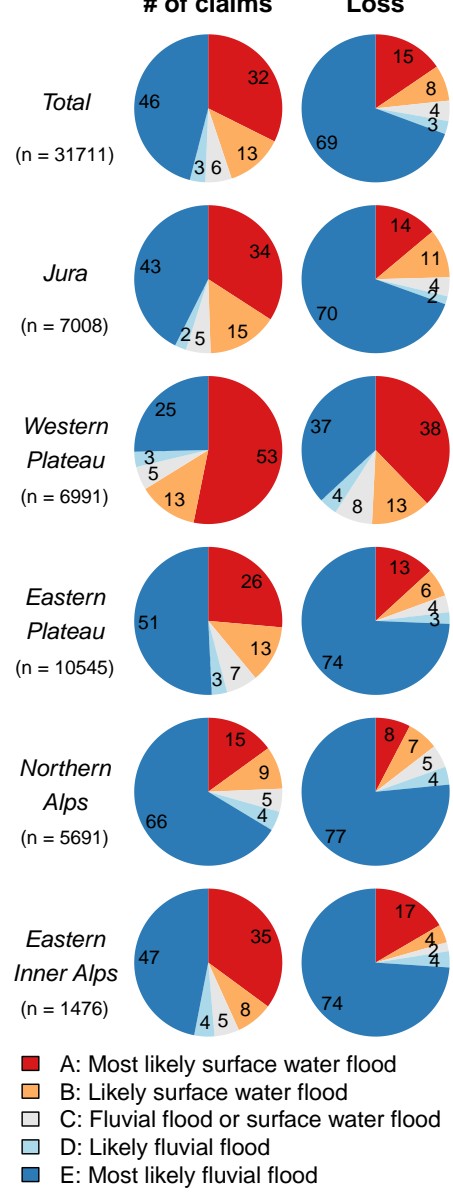

**Figure 8.** Number of claims as well as corresponding losses in total, and separately for each region. The values stem from the data set *D2*, which contains seamless claim records of 13 PICB covering the period of 1999–2013 (c.f. Table 2). The numbers indicate the shares in %, while *n* represents the sample size.

– The largest events cause most of the losses, whereas small events only account for insignificant losses in comparison.

Figure 10 has hinted at the fact that each event causes SWF damages alongside fluvial flood damages, except a few of the smallest events. This is further explored by Fig. 11. For each event, i.e. a day with at least one flood damage of any class, the




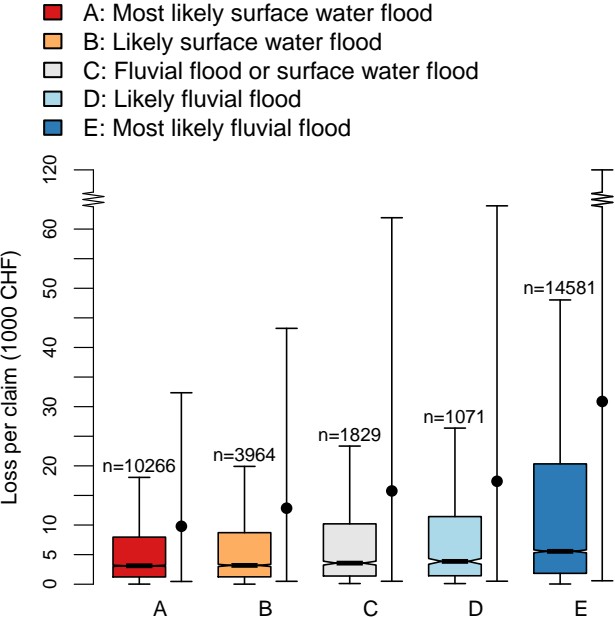

**Figure 9.** Box plots of losses per claim showing the interquartile range, i.e. the range from the 25 % to the 75 % percentile, as well as the median (bold horizontal line). Non-overlapping notches indicate significantly differing medians, while the whiskers extent to 1.5 times the interquartile range. Note that the outliers are not plotted. Instead, the 5–95 % percentile range is plotted on the right of each box plot, while the mean value is indicated by the solid dot. Furthermore, note that the y-axis is compressed between CHF 60'000 and 120'000. The plot is based on the data set *D2* (c.f. Table 2).

number of claims classified as SWFs is plotted against the number of claims classified as fluvial floods. As expected, most of the events are clustered around the origin, owed to the fact that events with up to 5 claims account for 74 % (1'100) of the total number of events (1'490) within the period of 1999–2013.

The most severe floods within the last 15 years within the study domain are highlighted in Fig. 11, which highlights that these flood events are also associated with high numbers of SWF damages, even though these events are mostly known for being devastating fluvial floods. Thus, our analyses show that fluvial flood damages generally coincide with SWF damages. This has been noted before (e.g. Blanc et al., 2012) and can be explained by the fact that both flood types are generated by the same rainfall input. Particularly, during extreme rainfall events, we can expect fluvial flood damages, as well as SWF damages. However, the shares of SWF damages in comparison to fluvial flood damages are different, which might be linked to the type of rainfall. For instance, the event on the 21–22 June 2007 was caused by high intensity rainfall (Hilker et al., 2008) and is associated with a larger share of SWF damages (Fig. 11). All other highlighted extreme flood events were triggered by long-duration rainfalls and, at the same time, larger numbers of fluvial flood damages. This could be an indication that the type of rainfall, and in particular the rainfall intensity, is an important driver of SWF damages, as noted for instance by Spekkers et al. (2013), as well.




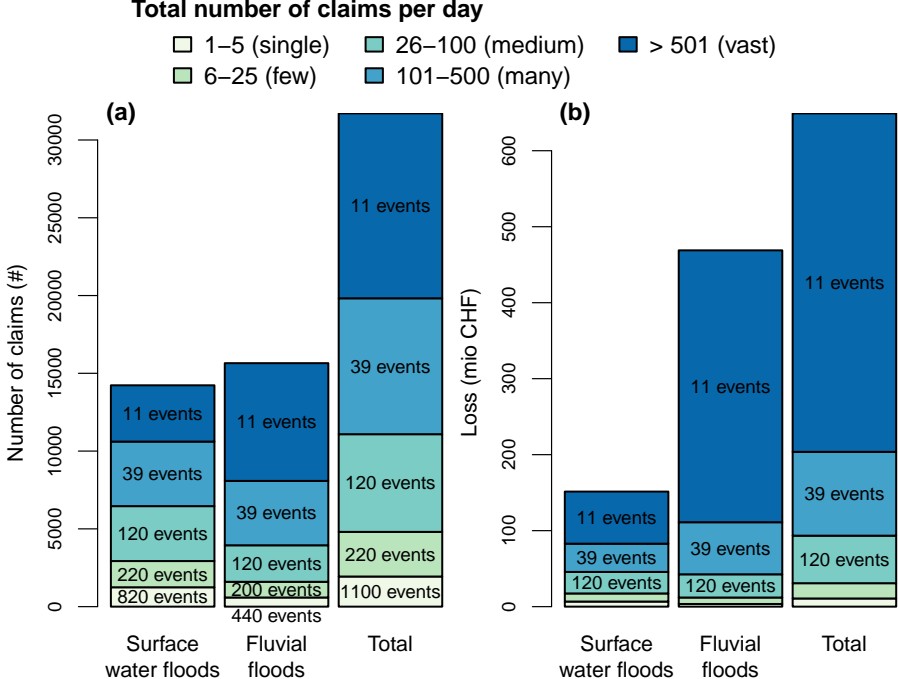

**Figure 10.** The total number of claims and loss categorized according to the size of the corresponding event, based on the data set *D2* (c.f. Table 2). **(a)** Each claim was categorized according to the total number of flood damages that occurred on the same day. For instance, all claims that occurred on 21 June 2007 fall into the category *vast*, since 1'162 damages were registered for that day in total. Thus, all these claims belong to one of the 11 largest events within the period of 1999–2013. As each claim was classified (Sect. 3.2), we can further group the data as claims related to SWFs (class A and B) or fluvial floods (class D and E), respectively. For the lowest two categories, i.e. *single* and *few*, the number of events of SWFs is larger than the number of fluvial flood events. This is due to the fact that some of these events consist of claims categorized as SWFs only. For all other categories, the event numbers match, indicating that for each of these days, some of the claims were classified as SWFs, while some were classified as fluvial floods. **(b)** The same stratification is applied to the associated loss. Note that the indication of the number of events for the smallest two event categories, i.e. *single* and *few*, were omitted for better readability. However, the values are identical to the values shown in panel a.

## 4.3   Spatial distribution

Thanks to the spatially explicit input data, we can get a good overview of SWF damages in space, as shown in Fig. 12. In general, it can be observed that the Swiss Plateau (2 and 3) is exposed most to SWFs, both in relative and absolute terms. Also in the Jura (1), many buildings are affected by SWFs. In contrast, the alpine regions of Switzerland, i.e. the Northern Alps (4) and also the Eastern Inner Alps (5) are exposed the least.

The visualization of the damage densities has advantages. For instance, in Bernet et al. (2016) low inundation rates by overland flow were reported for Grisons, i.e. the Eastern Inner Alps, and high values for Fribourg, which lies mostly in theWestern Plateau. Fig. 12 supports these findings, but presents a more differentiated picture, as differences within the mentioned regions


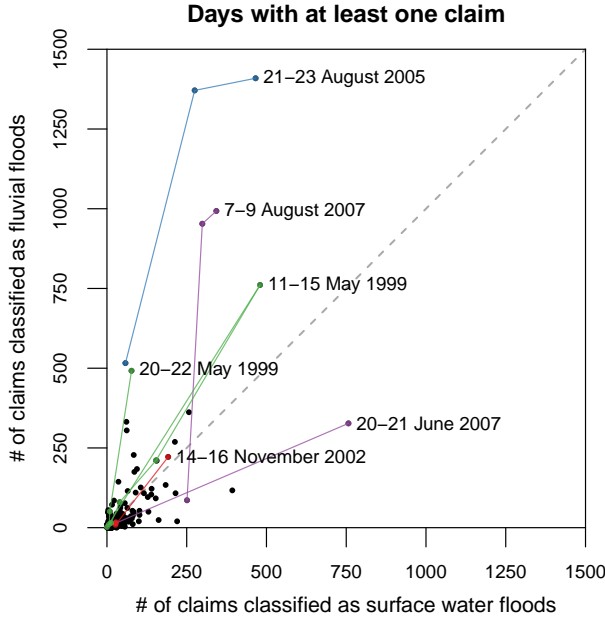

**Figure 11.** Scatterplot between the number of claims classified as SWFs (class A and B) against claims classified as fluvial floods (class D and E) based on the data set *D2* (c.f. Table 2). Each point represents an event, i.e. a day with at least one flood damage of any class. Along the dashed gray line, the number of SWF claims and fluvial flood claims are identical. Thus, claims below the line indicate events with more SWF than fluvial flood claims, and events above the line indicate the opposite. The severest flood events within the period of 1999–2013 are highlighted, in addition to the event in November 2002 that was the most significant event for the Eastern Inner Alps (c.f. Fig. 13). Thereby, all dates that belong to the same event are connected with lines, and severe events of the same year are shown in the same colors. The event dates are based on Hilker et al. (2008, 2009).

can be grasped, as well. In particular, we can see that the damage densities are not evenly distributed in space. The most affected regions are certainly those with high relative, as well as absolute number of damages, such as the areas indicated by the solid and dashed ellipses in Fig. 12. In addition, we see that such areas do not necessarily coincide with the most densely populated areas (dashed ellipse), but may lie in less populated areas (solid ellipses). Moreover, we can also identify areas that suffer from

5  high absolute number of damages but are exposed less in relative terms (dotted ellipse).

## 4.4 Temporal evolution

To obtain an idea about the distribution of the damages throughout the year, we have plotted the number of claims as well as associated losses against the month in which they occurred in form of spider plots (Fig. 13). In relation to SWFs, by far the most damages occur in the summer months from June to August in all regions, except in the Eastern Inner Alps. In the latter

10  region, the maximum number of damages were registered in November, which can be attributed to a single event that occurred on 14–16 November 2002 (Romang et al., 2004), which is highlighted in Fig. 11, as well. The remaining damages occurred




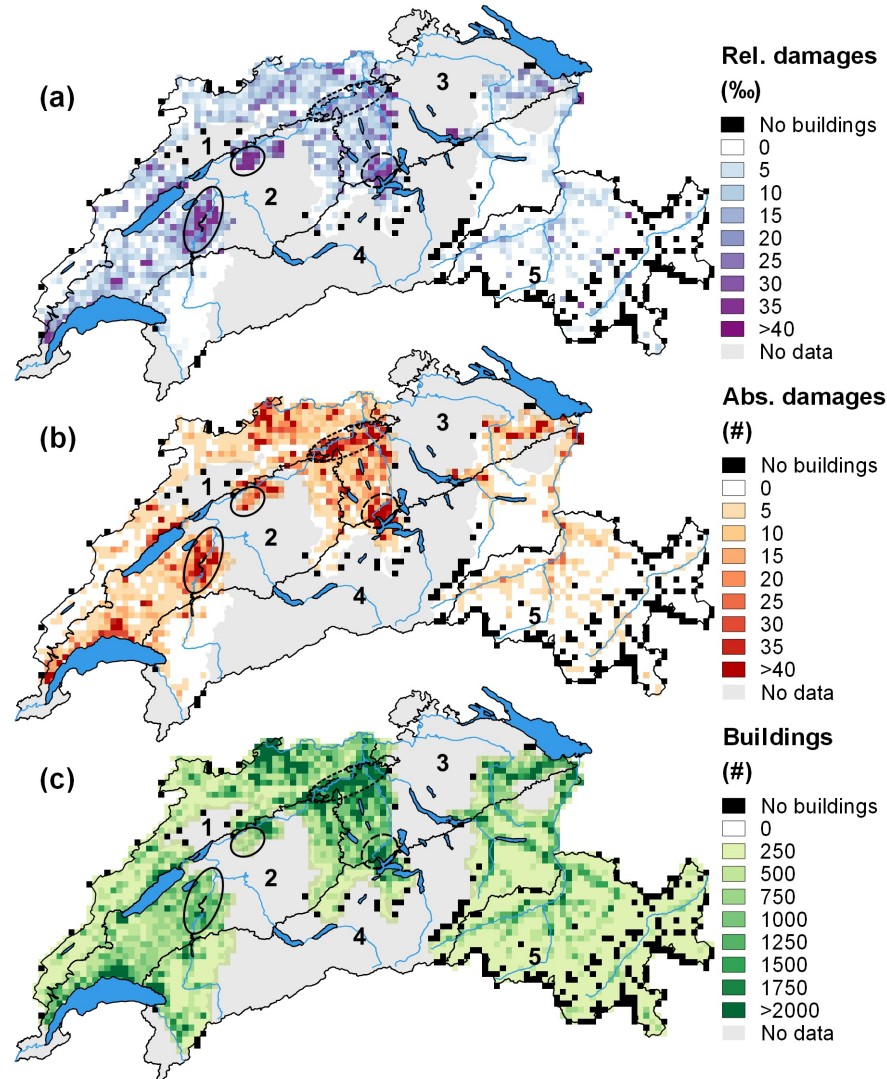

**Figure 12.** Relative (**a**) and absolute (**b**) numbers of damages caused by SWFs based on the data set *D2* covering the period of 1999–2013 (c.f. Table 2), aggregated to regular grids of 3 by 3 km. In addition, the absolute number of buildings per cell are shown (**c**). The solid ellipses highlight two less populated areas with high relative and absolute number of damages. The dashed ellipse indicates a highly populated area with high absolute and relative values, whereas the dotted ellipse marks a densely populated area with high absolute number of damages, but comparatively low relative values. The numbers indicate the corresponding region, i.e. Jura (1), Western Plateau (2), Eastern Plateau (3), Northern Alps (4) and Eastern Inner Alps (5).

also mainly in summer, but due to the devastating event in fall 2002, the values are much lower in comparison to the other regions.



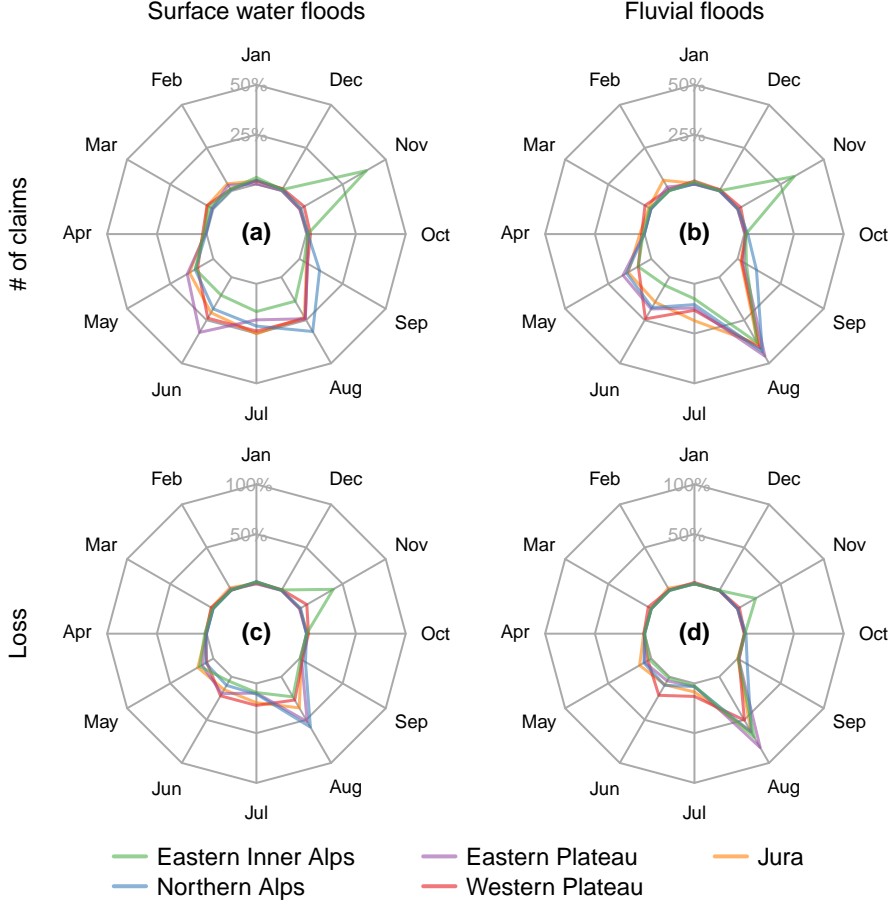

**Figure 13.** Spider plots indicating the relative number of damages and associated losses for each month. Separate plots are shown for SWFs (class A and B) and fluvial floods (class D and E). The data set *D2* constitutes the underlying data and covers the period from 1999–2013 (c.f. Table 2). Note that the scale for the number of claims ((**a**) and (**b**); 0–50 %) is not the same as the scale for the loss ((**c**) and (**d**); 0–100 %).

Overall, the number of claims are elevated in the last month in spring, i.e. May, and to a smaller degree in the first month of fall, i.e. September, for most regions. During the rest of the year, i.e. from October to April, very few damages occur, except for the Eastern Inner Alps in November, as discussed before.

Analogous to the number of damages, SWFs cause most of the associated losses in the summer months (Fig. 13c). Interest-
5   ingly, the losses in the Eastern Plateau and the Northern Alps have larger shares in August, compared to the other regions, but also compared to the corresponding number of claims (Fig. 13a). This can be explained by the particularly high losses during the August 2005 flooding, as indicated in Fig. 12.

The number of claims and associated losses of fluvial floods are highly concentrated in August in all regions (Fig. 13b and 13d). The event in November 2002 that affected the Eastern Inner Alps is also showing up prominently for fluvial floods, as
10   elaborated before.



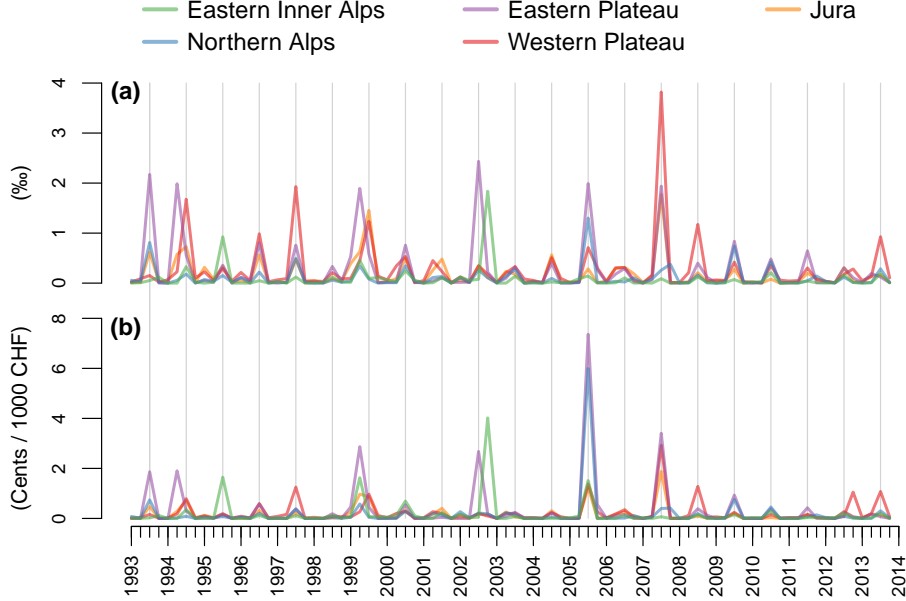

**Figure 14.** Time series showing the normalized number of SWF damages (**a**), as well as associated loss (**b**), based on the data set *D1* (c.f. Table 2). As pointed out in Sect. 3.3, not all data records cover the whole period, thus the representativeness is decreasing starting from 2003 as we move back to 1993. Nevertheless, as the aggregated values match well with the reference values (c.f. Fig. 7), and only relative values are considered here, the values are still meaningful.

Finally, it is interesting to have a look at the time series of damages caused by SWFs. Based on the normalized values covering the period of 1993–2013, we are able to show the relative number of claims and losses related to SWFs, individually for each region (Fig. 14). The seasonally aggregated values show a distinct pattern. The relative number of claims were almost always highest during the summer, i.e. in June, July and August, which supports the results discussed before. However, there are a few exceptions such as the spring of 1994 and 1999, where corresponding values exceeded the highest values of the same year. Interestingly, in both cases high values were also observed in the following summer, but in other regions. In 2002, a high value in summer that affected the Eastern Plateau was followed by the severe damages in the Eastern Inner Alps in November 2002 (Romang et al., 2004), which corresponded to the highest observed value in that region during the whole studied period. High values occurred frequently in the Eastern Plateau, but also in the Western Plateau, where in 2007 almost 4 ‰ of claims per buildings were registered. The highest values in the Jura occurred in summer 1999 and 2005. A value higher than 1 ‰ was observed in the Northern Alps only once, namely in 2005.

The values in terms of loss are in line with the claims per buildings, however, they are scaled differently. Most pronounced are certainly the high values in the Eastern Plateau and the Northern Alps in 2005. Other high values are observed in spring 1999, summer and fall 2002 as well as in summer 2007.

Furthermore, the data do not exhibit any trends of SWF damages in the period of 1993–2013 based on the seasonal Mann–Kendall test at a significance level of 0.1, except for the Jura. In that region, the number of claims have been decreasing





($p = 0.006$). In contrast, the relative losses in Jura do not exhibit such a trend ($p = 0.52$). The absence of any increasing trend might be a surprising result, as increasing damage trends are often reported (e.g., Kron et al., 2012; Grahn and Nyberg, 2017). However, it is important to note that in this study we are talking about normalized, relative values, while in the aforementioned publications, the trends of the absolute numbers are considered.

## 5 Discussions

Our results show that SWFs caused almost the same number of damages as fluvial floods. For the first time, these numbers are based on a large data set including more than 30'000 damage claims covering 15 years and 48 % of all Swiss buildings. Notably, SWFs only account for roughly one-quarter of the total loss, which is inline with results from the pilot study (i.e. Bernet et al., 2016). Nevertheless, the associated yearly loss is highly significant, as the following numbers exemplify: The

median of total yearly losses to buildings caused by fluvial floods within the considered regions is even slightly lower (5.0 mio CHF $y^{-1}$) than the median of SWF damages (5.9 mio CHF $y^{-1}$) based on the data set *D2* covering the period of 1999–2013 (c.f. Table 2). However, the mean yearly loss of fluvial floods is more than 3 times the loss caused by SWFs (i.e. 31.3 mio CHF $y^{-1}$ versus 10.1 mio CHF $y^{-1}$, respectively). The difference between the maximum yearly losses caused by each flood type is even more pronounced: While the maximum loss of SWFs amounts to 38.3 mio CHF $y^{-1}$ in 2005, fluvial floods caused 234.3 mio

CHF $y^{-1}$ in the same year, which corresponds to a factor of roughly 6.

These observation concerning annual flood losses are supported by the characteristics of the individual losses. Their exploration (Fig. 9) expressed that the range of loss per claim is much narrower for SWFs than for fluvial floods. As SWFs are expected to be associated with significantly lower flow depth than fluvial floods, this might be one of the main reasons for the lower associated loss, since water depth is among the most significant single impact parameters for structural damages to

residential buildings (e.g. Kreibich et al., 2009; Merz et al., 2013). Interestingly, the median loss of each claim associated with fluvial floods is also rather low, although significantly higher than the median loss of claims related to SWF. Yet, the highest losses per claim are caused by fluvial floods during the most severe events within the study domain (Fig. 10). As during extreme events, larger areas are affected and the associated shares of objects inundated by large water depths are higher (Elmer et al., 2010), higher losses per claim can be expected. Along the same lines, Hilker et al. (2009) report that the most severe events

contribute to more than half of the estimated total loss and Barredo (2009) found an even higher share for flood losses in the whole of Europe. Undoubtedly, loss ratios are higher during more extreme events (Elmer et al., 2010). Although, this probably also holds true for damages caused by SWFs, such damages certainly seem less influenced by the severity of the event (c.f. Fig. 9 and 10). Consequently, SWFs may rarely cause the total destruction of a building, and associated loss ratios may, thus, mostly be well below 1.

As outlined in the introduction, this study is limited to direct tangible damages to buildings. Therefore, the absolute loss values are low in comparison to other loss estimations that include other losses, as well. For instance, Hilker et al. (2009) report a mean financial loss of 317.2 mio CHF $y^{-1}$ between 1972–2007, which is roughly 7 times higher than the mean of all flood losses to buildings, as represented by our data set. For one, the data published by Hilker et al. (2009) cover the whole of





Switzerland and consider a longer period. More importantly, however, these estimates also include damages to infrastructure, forestry and agricultural land, in addition to damages to buildings and their content. Therefore, the associated losses are inherently higher than the numbers presented in this study. This exemplifies that one has to be careful when comparing values from different data sources (Kron et al., 2012). Moreover, it highlights the fact that the damages to buildings are just a small
fraction of the total loss caused by SWFs for the society. Nevertheless, these data serve well for assessing the relevance of SWF damages in Switzerland, especially when considering relative values.

The spatial distribution of damages caused by SWFs can be deceiving: Obviously, an area with a higher building density will likely result in a larger number of damages compared to an area that is less populated (Fig. 12b versus 12c). Therefore, it is important to have a look at relative values, as well (Fig. 12a). Thereby, the effect of higher values caused by a denser
number of buildings is considered. However, the relative values are quite sensitive in sparsely populated areas. A damaged house with virtually no other houses in the vicinity will produce a high relative value or a low value, respectively, if the same is not affected. In contrast, in more populated areas, the relative value will not change much in case a building is more or less damaged. Thus, to obtain a complete picture, the relative and absolute values should be considered alongside the building density. In that way, the most exposed areas can be identified, like the two highlighted areas in the Western Plateau that are
associated with high relative and absolute numbers of damages (Fig. 12).

Furthermore, it is important to keep in mind that in case an area has no registered damages, it does not necessarily mean that the area has not been affected by any floods at all. It just indicates that either no buildings were in the vicinity of the flooded area or the buildings were properly protected against such floods. Therefore, damage records can only indicate floods that lead to some sort of damage and never to the occurrence of floods in the hydrological sense, as discussed in the introduction
(c.f. Fig. 1). However, understanding the characteristics of damaging floods can open the stage to understand the process in a broader context, as well.

The temporal distribution of claims related to SWFs exhibits a distinct seasonality (Fig. 13 and 14). Similar to the flood losses reported by Hilker et al. (2009), most damages clearly occur in summer, with a few exceptions. Thereby, thunderstorms associated with short but intense rainfall are certainly an important driver of SWF damages. Nevertheless, long duration rainfall
events are also responsible for a large share of SWF damages, highlighted by the most severe events that are mostly associated with long duration precipitation. In contrast, much fewer damages are caused in spring and fall, and virtually no damages are caused in winter. Damages in winter can likely be attributed to rather local events coinciding with conditions promoting overland flow generation such as rain on frozen soils. Overall, these observations have important implications for assessing the hazard of SWFs. In particular, simply focusing on high intensity rainfall events may lead to an underestimation of the risk of
SWFs.

Although the time series is relatively short, the data do not exhibit any increasing trends of SWF damages in the period of 1993–2013. Obviously, the general increase of absolute damages in time, which can be found in our data as well, is eliminated when the data are normalized. Thus, as suggested for instance by Kundzewicz et al. (2014), the increase in loss can be mainly attributed to the socio-economic development. However, we did not consider further aspects that could have an influence on
such trends, such as a change in vulnerability (Bouwer, 2011). Moreover, insurance or local governmental policies that might




have changed over time were not taken into account either. Nevertheless, it is important to note that increasing absolute losses are most likely not attributable to climate change, but to socio-economic factors. Consequently, the major associated risks related to SWF damages is not climate change, but the increased exposure due to population growth and increasing wealth. This has implications for decision and policy makers, as well as for insurance companies and similar stakeholders.

The key to the exploitation of the insurance data with regards to SWFs lies in the classification of the damage claims (beside the provision of the data in the first place). The classification scheme, as introduced in this study, is based on the geographical location of each damage with respect to known fluvial flood zones and the hydrological network. On the one hand, this obviously requires spatially explicit damage data. On the other hand, it provides a reproducible, objective and most importantly, an independent classification. These characteristics are important, as the following examples highlight: Grahn and

Nyberg (2017) had to exclude damages that occurred on the same day as known fluvial floods in order to distinguish pluvial from fluvial flood claims. However, our results show that fluvial flood damages almost always coincide with damages caused by SWFs. Consequently, excluding damages occurring on the same day as fluvial floods likely introduces a bias. Another example is the statistical model applied by Spekkers et al. (2013) in order to differentiate rainfall-related damages clustered around wet days from non-rainfall related damages occurring throughout the year. Thereby, the classification of each claim is

not independent anymore, but is depends on how many other damages occurred on the same day.

    Although, the classification scheme presented in this study has striking advantages, it has the following short-coming: As overland flow propagates over the land surface, it may eventually reach a watercourse. Areas alongside watercourses, where the overland flow joins the river, may be a flood hazard zone. If so, all claims in that specific area are classified as a fluvial flood, even though, possibly, the claim might have been caused by incoming overland flow. However, a more qualified classification

entails likely event-specific, time-consuming manual assessments. In fact, it is extremely difficult to disentangle the different flood types, even more so for events in the more distant past and if no data with that particular focus are available. In contrast, for claims that are located far away of any watercourse, it is very unlikely that they are affected by watercourses at all. Therefore, our methodology renders a lower boundary of claims associated with SWFs, in essence. In reality, the numbers are likely higher, but as mentioned before, disentangling the flood types within their overlapping domains is difficult.

Indeed, the flood processes are a complex interlinked system, as Evans et al. (2004) stated. In fact, the insurance data illustrated that damages caused by SWFs occur (almost) always alongside claims caused by fluvial floods (c.f. Fig. 11). Be it a short and intense thunderstorm or a long duration event, rainfall is the main trigger of every SWF, as well as (almost) every fluvial flood. Understandably, if there is enough rainfall to cause a SWF, it may as well cause or at least contribute to a fluvial flood, once part of the water reaches the next watercourse. Undoubtedly, severe events that include hundreds or thousands of

damages entail a combination of flood processes, while, of course, some local events may be associated with a single flood process only.





## 6 Conclusions

In this study, we have presented a simple and pragmatic approach of how spatially explicit insurance data records can be exploited to investigate damages caused by SWFs. The methodology provides a robust lower estimate of SWF damages. Using the presented percentile values (Table 3), the methodology is applicable for classifying any claim in Switzerland, except in the Western Inner Alps and Southern Alps, where data were lacking. For these regions, appropriate values could be approximated. Moreover, the methodology is transferable to other regions and countries, but has to be adapted to locally available flood hazard maps.

There seems to be a consensus among Swiss practitioners and experts that SWFs are responsible for a large share of all flood damages. However, this perception does not stem from quantitative research, but rather from single case studies or practical experience. With the study at hand, we are able to quantify the striking relevance of SWFs in Switzerland based on a sound data basis, regionally representing 39–100 % of all buildings over a period of 15 years. The data reveal that SWFs cause nearly as many damages as fluvial floods. In contrast, SWFs account for roughly one-quarter of the direct tangible losses, driven by lower losses per SWF claim. This hints at the different processes' characteristics with generally low flow depths associated with SWFs, opposed to both low and high, static and dynamic, flow depths during fluvial floods that are additionally sensitive to the severity of an event.

The most affected areas are clearly the Western Plateau, both in relative and absolute terms, followed by the Eastern Plateau and the Jura. The more mountainous regions, i.e. the Northern Alps and the Eastern Inner Alps, are affected less. Notably, there are also large differences between the spatial distribution of damages within each region. By relating the absolute number of damages to the number of buildings in the vicinity, the effect of varying building densities can be considered. Nevertheless, in sparsely populated areas the relative numbers are sensitive and, thus, less robust, owed to the particularly low building densities. Furthermore, not all regions are affected by SWFs to the same extent throughout the year. Yet, in all regions most of the damages occur in summer, safe a few exceptions.

In general, the spatial and temporal distribution of SWF damages is complex. Different factors might be responsible for high damages within certain areas or during certain periods. For instance, the meteorological forcing differ spatially and temporally, the predisposition due to unfavorable soils or landuse practices play a role, past human interventions such as the installation of drainage and the removal of small natural rivulets can have an influence, but also slightly differing practices by the insurance companies or different rules applied for buildings to be built might be relevant. Undoubtedly, we stand at the beginning of better understanding SWFs in Switzerland, and also on an international level. Thereby, a common terminology is the base to strengthening and extending the science within this field across the countries' borders.

This study highlights the fact that SWFs are a highly significant flood process in Switzerland. Unlike for fluvial flood hazards, there is no publicly available information about the hazard of SWFs up to date, in spite of the process's obvious relevance. Since SWFs can occur practically anywhere in the landscape, the more paramount it seems to have detailed information about local SWF hazards. Such information can help to make well-founded decisions by all different stakeholders, e.g. planning and installing appropriate property protections by house owners, applying measures to reduce overland flow generation on




agricultural fields by local farmers, providing surface retention ponds by municipalities or amend regulations to prevent SWF damages by the federal government. However, as a first priority, SWFs in general and the influencing factors of SWFs in particular should be further studied, and, ultimately, better understood.

As a first step in this direction, we propose that SWFs should not be regarded as an isolated process by itself. A better way
is probably to extend our focus from rivers and lakes alone to hidden rivulets, covered drains, the sewer system, impervious areas, agricultural fields and headwaters, which all contribute to the generation of SWFs. Therefore, we should regard overland flow and ponding as an integrated part of our catchments. In this manner we may start to understand the complex interlinked flood processes better in the future.

## 7   Data availability

The data, on which this study is based, were provided by 15 different insurance companies. Each record contains confidential information such as the location (address and/or coordinates), claim date, associated loss etc. Due to privacy protection, the data are subjected to strict confidentiality and, thus, cannot be made accessible.

## Appendix A: Normalization

Obviously, it would be best to normalize the damage data with the corresponding property data of the respective insurance
company. However, property data are generally even more difficult to obtain than damage data, as the former contain additional sensitive and confidential information. Therefore, ancillary data are required to estimate the number of insured buildings, as well as the replacement value of each corresponding building. Moreover, as these values change over time, we need additional ancillary data to take these temporal changes into account. As outlined in Sect. 3.3, the spatial normalization as well as the temporal normalization of the damage data are described in detail in the following sections.

After all, we divide the number of claims by the estimated number of insured buildings, while the losses are divided by the corresponding total sum insured, respectively. For that matter, all quantities have to be spatially aggregated. For this study, we aggregated the data to regular grids and tested various resolutions. We chose a resolution of 3 by 3 km, which seemed like a good compromise between level of detail and smoothing. The point of origin of the corresponding rasters is chosen arbitrarily. Thereby, we note that the choice may change the absolute values of each cell, but in general does not change the larger picture.

## A1   Spatial normalization

As property data were not available, we inferred the number of buildings using ancillary data. For this purpose, we made use of the terrain model swissTLM3D (Table 1). From this data set, the number of buildings represented by their footprints can easily be extracted. However, the data needed to be preprocessed: Invalid geometries had to be corrected and overlapping polygons were dissolved into single polygons in order to obtain a homogeneous data set as of 2013.


The definitions of a building are quite similar among the PICB (Imhof, 2011). Nevertheless, the number of footprints does not match the number of insured buildings, since a row house might be represented by one footprint, while it constitutes several buildings as defined by the respective insurance company, for instance. To consider this, we referred to publicly available annual reports of 2013 and thereby obtained the total number of insured buildings for each PICB. We then divided the obtained values

by the number of footprints, resulting in a simple multiplication factor ($f_n$, Table A1). By multiplying the aggregated number of buildings with the factor $f_n$, we obtain the approximated number of insured buildings as of 2013. For each grid cell, the aggregated number of claims is then divided by the aggregated number of buildings to obtain spatially normalized damage numbers.

To normalize the loss, we need to relate the loss values to the total sum insured. There are few published methodologies

to assess building values in detail, but these can be too time-consuming for applications in large study areas (Kleist et al., 2006). Given the large data set, we chose a simple approach similar to the methodology shown by Grünthal et al. (2006), who used the product of mean insurance values and the number of buildings to estimate the replacement costs of residential buildings. However, instead of the buildings' footprint area, we considered the buildings' volume, which we expect to be a more representative measure for estimating building values.

Specifically, we first assessed the mean altitude of each building's footprint by using common zonal statistic functions of a GIS as well as a digital elevation model as input (Table 1). The top of each building was then assessed by the same methodology, but using a digital surface model, instead. The approximated building height resulted from the difference of the two values. Implausible results were corrected, i.e. values below 3.5 m or above 100 m were set to the standard building height of 3.5 m. Thus, a standard height of 3.5 m is assigned for buildings that might have been built after the last update of the DSM in 2008

(c.f. Table 1). Following, the building volumes are obtained by multiplying the building's footprint area with the mean building height. The total building volume for each canton is assessed and divided by the respective total sum insured, in order to obtain the insurance value per cubic meter ($\rho_v$, Table A1). The product of each building's volume and $\rho_v$ finally results in each building's value as of 2013. Analogous to the number of buildings, the loss is aggregated to regular grids and divided by the aggregated sum insured.

**A2    Temporal normalization**

As the considered terrain model itself does not include attributes for such considerations, we used another auxiliary data set, i.e the buildings and dwellings statistic of the Swiss Federal Statistical Office as of 2013, from which the number of newly built residential buildings can be inferred (Table 1). The data are regularly updated, whereas the number of residential buildings can be assessed at any time by linear interpolation between the sampling points. Normalizing with the number of

buildings per canton as of 2013 (Appendix A1), we obtain a dimensionless factor ($f_t$, Table A2). With the assumption that the residential buildings are representative for the development of all buildings, we obtain the temporal development of the number of buildings and the total sum insured. To that end, we multiply the interpolated factor $f_t$ for each time step with the number of buildings and the total sum insured as per 2013, respectively.





*Competing interests.*   The authors declare that they have no conflict of interest.

*Acknowledgements.*   Funding from the Swiss Mobiliar supported the completion of this research. We thank the Swiss Mobiliar in general and the natural hazards group in particular for acquiring and compiling the flood hazard maps as well as providing claim records. Furthermore, we would like to thank the public insurance companies for buildings of the cantons Aargau, Basel-Landschaft, Basel-Stadt, Fribourg, Glarus,

5   Grisons, Jura, Lucerne, Neuchâtel, Nidwalden, St Gall, Solothurn, Vaud and Zug for providing claim records and supporting us during the data harmonization process. Also, we would like to thank the canton of Lucerne for providing the overland flow map. Last but not least, we thank Markus Mosimann for his support of harmonizing the insurance data, Veronika Röthlisberger for the estimation of the buildings' values as well as her and Andreas Zischg for the many valuable inputs.





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



**Table 1.** Summary of the specific input data used for the classification and normalization of the flood damage claims, in the order of appearance in Fig. 2. Note that all links were last checked on 3 March 2017.

| Input data | Name | Description | source |
|---|---|---|---|
| Address data base | GeoPost Coordinates | Register of all geocoded postal addresses of Switzerland as of 2015, provided by the national postal service Swiss Post | *https://www.post.ch/en/ business/a-z-of-subjects/ maintaining-addresses-and-using-geodata/ address-and-geodata* |
| River network | swissTLM3D | Feature TLM_FLIESSGEWAESSER of the Swiss topographical landscape model, v1.4, provided by the Federal Office of Topography (swisstopo) | *https://shop.swisstopo.admin.ch/en/products/ landscape/tlm3D* |
| Flood hazard maps (main) | Flood hazard maps | Official Swiss (fluvial) flood hazard maps (e.g., Zimmermann et al., 2005; de Moel et al., 2009) compiled to a single data set and provided by the Swiss Mobiliar | *https://www.bafu.admin.ch/bafu/en/home/ topics/natural-hazards/state/maps.html* |
| Flood map (ancillary) | Aquaprotect | Simple flood map for the whole of Switzerland, produced by the Swiss Federal Office for the Environment (FOEN) in collaboration with the Swiss reinsurance company Swiss Re | *https://www.bafu.admin.ch/bafu/en/home/ state/data/geodata/natural-hazards--geodata. html* |
| Building footprints | swissTLM3D | Feature TLM_GEBAEUDE_FOOTPRINT, see river network for details | see river network |
| Digital elevation model | swissALTI3D | High precision digital elevation model (DEM) as of 2013 with a regular grid size of 2 by 2 m, provided by swisstopo | *https://shop.swisstopo.admin.ch/en/products/ height_models/alti3D* |
| Digital surface model | DSM | Digital surface model, last updated in 2008, provided by swisstopo | *https://shop.swisstopo.admin.ch/en/products/ height_models/DOM* |
| Residential buildings | BDS | Buildings and dwellings statistic, as of 2013, provided by the Swiss Federal Statistical Office | *https://www.bfs.admin.ch/bfs/en/home/ statistics/construction-housing/surveys/ gws2009.assetdetail.8945.html* |
| PICB characteristics | – | Total number of insured buildings and total sum insured of each considered PICB as of the end of 2013, taken from their annual reports | available online for most PICB |



**Table 2.** Characterization of the claim records reporting flood damages to buildings provided by 14 different PICB in addition to claims of content and buildings provided by the cooperative Swiss Mobiliar. The absolute number of localized claims is presented in addition to the fraction relating to the total number of claims. The columns *D0*, *D1*, *D2* each represents a data (sub-) set and indicates the temporal coverage of each data record (*D0*) or a specific limitation thereof (*D1* and *D2*), respectively.

| Company | canton | Localized claims | *D0* | *D1* | *D2* |
|---|---|---|---|---|---|
| PICB$_1$ | Solothurn (SO) | 4'456 (90 %) | 1981–2013 | 1993–2013 | 1999–2013 |
| PICB$_2$ | Glarus (GL) | 463 (56 %) | 1982–2013 | 1993–2013 | 1999–2013 |
| PICB$_3$ | Fribourg (FR) | 5'494 (96 %) | 1983–2013 | 1993–2013 | 1999–2013 |
| PICB$_4$ | Nidwalden (NW) | 1'383 (97 %) | 1987–2013 | 1993–2013 | 1999–2013 |
| PICB$_5$ | Neuchâtel (NE) | 1'959 (99 %) | 1988–2013 | 1993–2013 | 1999–2013 |
| PICB$_6$ | Aargau (AG) | 9'024 (73 %) | 1989–2013 | 1993–2013 | 1999–2013 |
| PICB$_7$ | Grisons (GR) | 2'258 (95 %) | 1991–2013 | 1993–2013 | 1999–2013 |
| PICB$_8$ | Basel-Stadt (BS) | 243 (86 %) | 1992–2013 | 1993–2013 | 1999–2013 |
| PICB$_9$ | Lucerne (LU) | 7'848 (79 %) | 1993–2013 | 1993–2013 | 1999–2013 |
| PICB$_{10}$ | Vaud (VD) | 3'275 (56 %) | 1994–2013 | 1994–2013 | 1999–2013 |
| PICB$_{11}$ | Basel-Landschaft (BL) | 1'820 (89 %) | 1999–2013 | 1999–2013 | 1999–2013 |
| PICB$_{12}$ | Jura (JU) | 809 (83 %) | 1999–2013 | 1999–2013 | 1999–2013 |
| PICB$_{13}$ | St Gall (SG) | 4'764 (74 %) | 1999–2013 | 1999–2013 | 1999–2013 |
| PICB$_{14}$ | Zug (ZG) | 761 (85 %) | 2004–2013 | 2004–2013 | |
| Swiss Mobiliar | All (build. & cont.) | 18'560 (100 %) | 2004–2014 | | |
| Total | | 63'117 (85 %) | 63'117 | 40'233 | 31'711 |




**Table 3.** Percentile values ($d_X$, whereas $X$ stands for the $Xth$ percentile) obtained from the ECDF of ACEDs between claims within flood zones and the closest river for each respective region. The column $n$ represents the sample size of each underlying ECDF.

| Region | $n$ (#) | $d_{25}$ (m) | $d_{50}$ (m) | $d_{75}$ (m) | $d_{99}$ (m) |
|---|---|---|---|---|---|
| Jura | 5'508 | 58 | 135 | 315 | 1'360 |
| Western Plateau | 5'810 | 47 | 108 | 237 | 1'259 |
| Eastern Plateau | 10'167 | 56 | 135 | 285 | 1'084 |
| Northern Alps | 7'532 | 65 | 137 | 298 | 1'198 |
| Eastern Inner Alps | 891 | 29 | 61 | 112 | 643 |



**Table A1.** Factors used for the data normalization, i.e. the dimensionless multiplication factor ($f_n$) relating the number of building footprints to the number of buildings as defined by each PICB, as well as the estimated insurance value per cubic meter ($\rho_v$). For the derivation of these factors, the number of buildings and the total sum insured was required for each PICB. The corresponding values are generally published in the publicly available annual reports. Specifically, the values from the year 2013 were extracted for the PICB of the cantons of Aargau (AG), Basel-Landschaft (BL), Basel-Stadt (BS), Fribourg (FR), Grisons (GR), Jura (JU), Neuchâtel (NE), St Gall (SG), Solothurn (SO), Vaud (VD) and Zug (ZG). The values for the PICB of Glarus (GL) and Nidwalden (NW) were not reported, so that the mean value of 1.37 was adopted for the multiplication factor $f_n$ and the total sum insured was inferred indirectly from the respective annual reports.

|  | AG | BL | BS | FR | GL | GR | JU | LU | NE | NW | SG | SO | VD | ZG |
|---|---|---|---|---|---|---|---|---|---|---|---|---|---|---|
| $f_n$ (-) | 1.34 | 1.56 | 3.92 | 1.30 | 1.37 | 1.46 | 1.18 | 1.27 | 1.28 | 1.37 | 1.25 | 1.33 | 1.33 | 1.28 |
| $\rho_v$ (CHF m$^{-3}$) | 720 | 734 | 1056 | 577 | 582 | 868 | 449 | 575 | 656 | 706 | 628 | 664 | 733 | 1'029 |





**Table A2.** Multiplicative factor ($f_t$) indicating the number of buildings in relation to the total number of buildings as of 2013. In this table, the factor's values at the sampling points of the source data are shown (c.f. Table 1).

| Year | AG | BL | BS | FR | GL | GR | JU | LU | NE | NW | SG | SO | VD | ZG |
|------|------|------|------|------|------|------|------|------|------|------|------|------|------|------|
| 1980 | 0.58 | 0.63 | 0.91 | 0.53 | 0.78 | 0.66 | 0.70 | 0.56 | 0.74 | 0.61 | 0.65 | 0.63 | 0.67 | 0.55 |
| 1985 | 0.64 | 0.69 | 0.93 | 0.59 | 0.81 | 0.71 | 0.74 | 0.63 | 0.78 | 0.68 | 0.70 | 0.69 | 0.72 | 0.62 |
| 1990 | 0.72 | 0.76 | 0.94 | 0.67 | 0.86 | 0.79 | 0.80 | 0.70 | 0.84 | 0.75 | 0.77 | 0.76 | 0.79 | 0.70 |
| 1995 | 0.78 | 0.81 | 0.96 | 0.73 | 0.90 | 0.84 | 0.85 | 0.77 | 0.87 | 0.80 | 0.82 | 0.82 | 0.82 | 0.76 |
| 2000 | 0.85 | 0.88 | 0.97 | 0.80 | 0.94 | 0.89 | 0.89 | 0.84 | 0.91 | 0.86 | 0.88 | 0.88 | 0.86 | 0.84 |
| 2005 | 0.91 | 0.93 | 0.99 | 0.87 | 0.96 | 0.93 | 0.93 | 0.90 | 0.94 | 0.92 | 0.92 | 0.93 | 0.91 | 0.91 |
| 2010 | 0.97 | 0.98 | 1.00 | 0.95 | 0.99 | 0.97 | 0.98 | 0.96 | 0.98 | 0.97 | 0.97 | 0.97 | 0.97 | 0.97 |
| 2013 | 1.00 | 1.00 | 1.00 | 1.00 | 1.00 | 1.00 | 1.00 | 1.00 | 1.00 | 1.00 | 1.00 | 1.00 | 1.00 | 1.00 |