# Peer review of "Surface water floods in Switzerland: what insurance claim records tell us about the caused damage in space and time"

_Natural Hazards and Earth System Sciences, 2017_

## Referee Comment (RC1) · Anonymous Referee #1 · 22 May 2017

GENERAL REMARKS: 1. The paper discusses a very interesting theme as surface water floods are felt as an increasing problem in intensively settled areas more and more and demand for research on this theme is increasing. Thus, the paper gives significant contribution on objectifying the discussion on losses due to natural hazards like surface water flow (SWF) and matches the scope of NHESS. 2. In addition the paper presents a new way to interpret insurance data in terms of natural hazards. 3. Tools, methods, results are up to international standards. 4. Results of the investigations strongly support the interpretation and the conclusions drawn in the paper. 5. Desicription of data and methods used is sufficient except three points (pages 3, 9 and 12, see further comments below). 6. The title clearly describes the main concern of

the paper 7. Title and abstract are clearly presented and easy to read 8. Connex to previous and related research is sufficently described in chapter 1 9. Number of references should be enlarged in chapter 5. The results of the presented project should be discussed more in context of supranational and international studies (see further comments below) 10. Presentation of the paper is structured well and clearly presented except parts of chapters 2 and five (see further comments below) 11. Length of the paper seems adequate 12. Figures and tables are clearly presented, except figures 12 and 14 (some symbols are hard to distinguish) 13. Technical and english language are of good quality, fluently presented and understandable for an interdisciplinary audience

FURTHER COMMENTS: Ad chapter Introduction Gives a good overview about the actual situation. Page 2, line 16-17: "In order to reduce the risk, it is suggested to focus on the physical protection of exposed objects (e.g., Kron, 2009; DWA, 2013)…." Remark: This is one point of concern. But one step before: Adaptation and improvement of spatial management (hazard zone mapping, spatial development / management plans etc.) to reduce construction activities or to allow building activities only with special obligations in areas prone to SWF is necessary.

Ad chapter 2 – Terminology Presented in a more descriptive manner. From the announcement in the last paragraph of chapter 1 (page 3) a short and precise presentation of the most important terms, a precise definition for every term has been expected. Thus, presentation in form of a table with precise definionitions would be useful, in addition the chapter could be shortened and kept more precise. Page 5, Figure 1: A title / precise description is missing – what is Fig.1 showing? Page 5, lines 15-17, page 6, lines 1-5: No contribution of this text to improve systematic understanding. When it is the authors' opinion that these explanations are absolutely necessary – put them at the beginning of this chapter.

Ad Chapter 3 – Material and methods Page 6, line 31: "The provided data provided by…" – improvement of wording necessary Page 8, line 6-7: "…obvious errors were corrected, if possible, or removed otherwise…". Suggestion for further specification:

"…obvious errors were corrected, if possible, or the invalid data removed otherwise…“ Page 9, line 16: "…Thereby, the question ist how these distance…“ Suggestion: "…Thereby, the question is how this distance…“ Page 9, lines 30-31: "…To take regional geographical characteristics into account, the percentiles are calculated for each region separately (Table 3)…“. Question: How have special geographical characteristics been taken into account for the different regions. Give a short description or a reference where the procedure has been described alrady. Page 11, Figure 5: wrong: "…Intersects Aqua-portect flood zone…“ Suggestion: "…Intersects Aqua-protect flood zone…“ Page 11, lines 4-5: "…The smallest rivers were not considered by Aquaprotect, but there is no objective way of knowing where the cut-off was set…“ Question: Definition of the term "the smallest rivers“. A more precise information is necessary. Brooklets with l/sec, brooks with a few m$^3$/sec – which type of running water? Page 12, lines2-4: "…Nevertheless, the level of detail of the Swiss flood hazard maps far exceeds the one of Aquaprotect. This is taken into account by applying lower percentile levels for claims located within the flood hazard perimeters, as shown in Fig. 5 and 6…“ Question: On which basis of the lower percentile levels have been assessed? Systematic algorithm / empirical decision?

Ad chapter 4 – Results Page 13, lines 4-8: The passus "In Switzerland, there are few quantitative studies…..and covers only a small part of Switzerland.“ should be moved to chapter 5 – Discussion. Page 13, lines 30-31: "Nevertheless, the indicated hazard zones were mostly located in the vicinity of SWF claims.“ Improved precision of the statement necessary. Suggestion: "Nevertheless, the indicated hazard zones were mostly located in the vicinity of SWF claims, forming the data-basis of the investigations presented in this paper“ Page 17, line 10: "For instance, the event on the 21–22 June 2007 was caused by high intensity rainfall (Hilker et al., 2008)…“ Numerical data information desired, a range of the occurred rain intensities should be given (e.g. 5 mm h-1 over 30 hours, 200 mm d-1) Page 20, Figure 12: dashed, dotted and solid ellipses are very hard to distinguish Page 22, Figure 14: red lines (Jura and Western Plateau) are very hard to distinguish too

Chapter 5 – Discussion General remark: In this chapter the results of the presented project should be discussed more in context of supranational and international studies. The chapter lacks citations. So e.g. on Page 24, lines 26-30 and page 25, lines 1-4 (e.g. "Nevertheless, it is important to note that increasing absolute lossesare most likely not attributable to climate change, but to socio-economic factors...")

Page 25, lines 5-24: Discussing methodological issuses. Should be put at the beginning of chapter 5 - Discussion

Please also note the supplement to this comment:
http://www.nat-hazards-earth-syst-sci-discuss.net/nhess-2017-136/nhess-2017-136-RC1-supplement.pdf

---

## Referee Comment (RC2) · Anonymous Referee #2 · 20 Jun 2017

Using a comprehensive data set with flood insurance claims from Switzerland representing 48% of all buildings in Switzerland, the authors have developed an algorithm that allows them to assign a flood type (fluvial flood or surface water flooding - SWF) to each damage claim using official flood hazard maps as further input. After that, they present a thorough analysis of the data revealing the relevance of surface water flooding, their regional as well as their temporal distribution. Throughout the analysis the number of claims and the total amount of loss are distinguished. By this, interesting insights in damaging processes and the differences between the two flood types are revealed. Overall, the paper addresses an important field and provides unique data and insights on the relevance of surface warter flooding, on which a lot of rumours, but

only little evidence have existed so far.

Despite this overall positive assessment, I would like to mention a few suggestions how to further improve the paper. 1) For an international audience the flood insurance scheme in Switzerland should be introduced in more detail. 2) Section 2 should end with some recommendations on the usage of the terms/definitions or a clarification how they were used in the paper. In addition, the distinction between pluvial / SWF and fluvial floods should be explained. 3) The usage of the data from the Swiss Mobilar is not clear to me. On p. 6, line 7, it is stated that those data were not used, but later they are mentioned several times. Please clarify. In addition, I would recommend to avoid the company's name throughout the paper to avoid (unintended) promotion. 4) The section 4.1 on validation comes a bit as a surprise. I would prefer to read the methodological consideratiions and the inforamtion about data available for validation in the section on Data and Methods. The results on the validation need more explanations. 5) Since the results are presented in several sub-sections, the reader has to wait a long time for a discussion and interpretation of the findings. Therefore, I would recommend to have a common section on Results and Discussion and add some discussion at the end of each subsection. In addition, more explanations and interpretations of the findings would be great, e.g. by considering the different nature of the flood processes shown in Fig. 1, the spatial extent of triggering rainfall events and their seasonality. The advantages and shortcomings of the methodology could be shifted to the conclusions if this issue does not fit into the (new) more specific discussions. 6) There are a lot of figures; some could be improved in quality and readability, particularly Fig. 13 and 14. In addition, the style of Fig. 11 is a bit confusing and full. Maybe you should concentrate on the 11 events only.

The paper would also benefit from a language check.

Some minor points: - To my best knwoledge the plural of damage is not damages. Damages has a another meaning (in German: Schadensersatz/Entschädigung). - p. 2, line 2: Besides Amsterdam, the city of Münster, Germany, experienced heavy pluvial flooding on the same day causing one of the costliest hazard events for German insurers (see Spekkers et al. 2017 for a comparison of the event in the two cities); see: http://www.nat-hazards-earth-syst-sci-discuss.net/nhess-2017-125/ - p 2: The role and relevance of market penetration of flood insurance should also be addressed here. - p 3: Insurance data generally do not include further information about factors and processes that explain the amount of damage as this information is usually not recorded by insurers to save costs. This should be addressed in this section. - p. 3, line 33: The term "direct tangible flood losses" should be briefly defined. Add some information (maybe in the section on methods) on how you treated deductibles and upper bounds of insurance coverage (if applicable). - p.6, line 17-23: Fig 2 and Table 1 should be shifted to the section in which both are explained in detail. - p.8, line 7: Could you provide examples for data errors and their corrections? - p.8, line 8: To which index is the used index comparable? What are the underlying counted goods and services? - p.9, line 22-23: This process is not 100% clear. - p.10, line 1-2: In my view, this (lake inundations are covered by fluvial flood haazrd maps) is the most convincing point for the assumption made and should be mentioned earlier. - p.10, line 15: check terms (use hazard instead of danger) - p.12, line 20: Is there any evidence for this assumption? - p.15, line 22-23: Please add: how did you deal with consecutive days with claims when defining events? - p.18, line 6: damage density should be defined (and refer to the appendix explicitly) - p.26, line 8-9: The same is true for Germany, but the evidence is lacking. Maybe this point should be made clearer - already in the introduction. - p.27, line 23: Why? Please mention your assessment criteria.

I am looking forward to the revised paper. Thank you for this interesting analysis.

---

## Author Comment (AC1) · 19 Jul 2017

Final Response

>First of all, we would like to warmly thank the two anonymous referees as well as the editors for their time and effort to review this manuscript. We greatly appreciate their valuable and constructive inputs that help to improve the manuscript's quality.

>In the following, we go through each of the raised points by the two referees. Please note that this text is formatted for better readability and provided as a supplement.

General remarks of referee #1: 1. The paper discusses a very interesting theme as

surface water floods are felt as an increasing problem in intensively settled areas more and more and demand for research on this theme is increasing. Thus, the paper gives significant contribution on objectifying the discussion on losses due to natural hazards like surface water flow (SWF) and matches the scope of NHESS. 2. In addition the paper presents a new way to interpret insurance data in terms of natural hazards. 3. Tools, methods, results are up to international standards. 4. Results of the investigations strongly support the interpretation and the conclusions drawn in the paper. 5. Desicirion of data and methods used is sufficient except three points (pages 3, 9 and 12, see further comments below). 6. The title clearly describes the main concern of the paper 7. Title and abstract are clearly presented and easy to read 8. Connex to previous and related research is suffcently described in chapter 1 9. Number of references should be enlarged in chapter 5. The results of the presented project should be discussed more in context of supranational and international studies (see further comments below) 10. Presentation of the paper is structured well and clearly presented except parts of chapters 2 and five (see further comments below) 11. Length of the paper seems adequate 12. Figures and tables are clearly presented, except figures 12 and 14 (some symbols are hard to distinguish) 13. Technical and english language are of good quality, fluently presented and understandable for an interdisciplinary audience

>We greatly thank the referee for the time and effort to review our manuscript. We approve of most of the suggestions and propose changing the manuscript accordingly. The response to each specific point is listed in the following.

Specific comments of referee #1: Ad chapter Introduction Gives a good overview about the actual situation. Page 2, line 16-17: "In order to reduce the risk, it is suggested to focus on the physical protection of exposed objects (e.g., Kron, 2009; DWA, 2013). . .." Remark: This is one point of concern. But one step before: Adaptation and improvement of spatial management (hazard zone mapping, spatial development / management plans etc.) to reduce construction activities or to allow building activities only with special obligations in areas prone to SWF is necessary.

>We thank the referee for the valuable remark. We agree that the hazard of SWF should definitely be taken into account in spatial planning and should be managed accordingly. However, a prerequisite is that the hazardous process is understood well and can be predicted. In the cited documents, the authors argue that surface water floods (SWFs) can occur practically everywhere and are difficult to predict. Thus, they argue that the most effective measure is to focus on the physical protection of the object itself. In order to clarify that statement, we propose changing the citation as follows:

>In order to reduce the risk, an effective approach is to focus on the physical protection of exposed objects (e.g., Kron, 2009; DWA, 2013).

Ad chapter 2 – Terminology Presented in a more descriptive manner. From the announcement in the last paragraph of chapter 1 (page 3) a short and precise presentation of the most important terms, a precise definition for every term has been expected. Thus, presentation in form of a table with precise defionitions would be useful, in addition the chapter could be shortened and kept more precise.

>We acknowledge that a concise table summarizing the introduced terms might be helpful. If required, such a table could be introduced.

Page 5, Figure 1: A title / precise description is missing – what is Fig.1 showing?

>We agree that a title is missing. Figure 1 depicts the hydrological processes involved in surface water flooding and creates a link to the inherent differences between fluvial floods and surface water floods. While for fluvial floods, we experience the integral response of a river to a storm event, in the case of surface water floods we observe a response of the catchment before the excess water reaches the streamflow or a lake.

>We propose adding a precise description of the figure and revise its caption.

Page 5, lines 15-17, page 6, lines 1-5: No contribution of this text to improve systematic understanding. When it is the authors' opinion that these explanations are absolutely necessary – put them at the beginning of this chapter.
>We agree that the paragraph does not improve the systematic understanding of different terms. However, the paragraph was meant to give some possible explanation and background information about why the classification of flood processes is not simple. As such, we do not think that these explanations are indispensable and the passage could be moved or removed, if required.

Ad Chapter 3 – Material and methods Page 6, line 31: "The provided data provided by. . ." – improvement of wording necessary

>We thank the referee for pointing this out and suggest changing the wording as follows: The data provided by . . .

Page 8, line 6-7: ". . .obvious errors were corrected, if possible, or removed otherwise. . .". Suggestion for further specification: ". . .obvious errors were corrected, if possible, or the invalid data removed otherwise. . ."

>We acknowledge that this passage is not specific enough. In accordance with the second referee's request for some examples, we propose changing this passage as follows:

>During this procedure, the data were quality checked. Obvious errors such as address misspellings or flipped coordinate pairs were corrected. Furthermore, we removed duplicated entries, as well as records with incomplete (e.g. missing address) or invalid data (e.g. invalid damage date).

Page 9, line 16: ". . .Thereby, the question ist how these distance. . ." Suggestion: ". . .Thereby, the question is how this distance. . ."

>We thank the referee for pointing out the error and propose changing it as suggested.

Page 9, lines 30-31: ". . .To take regional geographical characteristics into account, the percentiles are calculated for each region separately (Table 3). . .". Question: How have special geographical characteristics been taken into account for the different regions. Give a short description or a reference where the procedure has been described alrady.

>We agree with the referee that this statement is not quite clear. The regions, by which the data are grouped and stratified, represent areas with similar geographical characteristics. Specifically, the regions are based on a classification of Switzerland into natural landscape units after Grosjean (1975). At the coarsest level of the presented classification, the author considered mainly orographic as well as climatic factors. As mentioned on page 6, lines 26-28, we adapted these regions, while constraining the regions' borders to the boundaries of hydrological catchments. Thus, by assessing the percentiles separately for each region, we implicitly take the geographical characteristics, by which the corresponding regions were classified, into account. To clarify the passage, we propose changing the text as follows:

>The percentiles are calculated for each region separately (Table 3). As the regions themselves represent areas with similar orographic and climatic characteristics (Grosjean, 1975), we thereby implicitly take these regional geographical characteristics into account.

>Furthermore, we realized that on page 6, lines 26-28, instead of "...adopted..." it should read "...adapted..." We propose applying this correction, as well.

Page 11, Figure 5: wrong: "...Intersects Aqua-portect flood zone..." Suggestion: "...Intersects Aqua-protect flood zone..."

>We propose changing the misspelling in Figure 5 according to the referee's comment.

Page 11, lines 4-5: "...The smallest rivers were not considered by Aquaprotect, but there is no objective way of knowing where the cut-off was set..." Question: Definition of the term "the smallest rivers". A more precise information is necessary. Brooklets with l/sec, brooks with a few $m^3$/sec – which type of running water?

>We agree that the expression "smallest river" is not appropriate. However, as mentioned in the corresponding passage, we could not find any information concerning a cut-off criterion or a method to obtain this information indirectly. The Aquaprotect's

documentation does not provide any indication concerning this issue. Thus, it remains unclear whether a minimum flow threshold, a minimum flow accumulation or catchment area or a similar criterion was considered. It is apparent, however, that Aquaprotect consistently does not cover headwaters. Similarly, small tributaries are not considered. However, it is not possible to infer and report a quantitative threshold from this data. Nevertheless, we suggest changing the passage as follows:

>Consistently, headwaters and small tributaries are not covered by Aquaprotect. Yet, no information about the specific exclusion criterion could neither be found nor inferred.

Page 12, lines2-4: "...Nevertheless, the level of detail of the Swiss flood hazard maps far exceeds the one of Aquaprotect. This is taken into account by applying lower percentile levels for claims located within the flood hazard perimeters, as shown in Fig. 5 and 6..." Question: On which basis of the lower percentile levels have been assessed? Systematic algorithm / empirical decision?

>We chose the lower percentile levels empirically, based on case studies. To reflect this decision, we propose changing the statement as follows:

>Nevertheless, the level of detail of the Swiss flood hazard maps far exceeds the one of Aquaprotect. We considered this by empirically choosing lower percentile levels for claims located within the flood hazard perimeters, as shown in Fig. 5 and 6.

Ad chapter 4 – Results Page 13, lines 4-8: The passus "In Switzerland, there are few quantitative studies.....and covers only a small part of Switzerland." should be moved to chapter 5 – Discussion.

>This passage was meant as an introduction to chapter 4 (Results), whereby the results to be presented are put into context. Given the rather stylistic purpose of this passage, we could move it to chapter 5 (Discussions), if desired.

Page 13, lines 30-31: "Nevertheless, the indicated hazard zones were mostly located in the vicinity of SWF claims." Improved precision of the statement necessary. Suggestion: "Nevertheless, the indicated hazard zones were mostly located in the vicinity of SWF claims, forming the data-basis of the investigations presented in this paper"

>We thank the referee for the suggested improvement. Also, the second referee requested a specification of this passage (see corresponding comment further below). The second referee proposed to move the whole chapter or at least parts thereof to chapter 3 (Materials and methods). Based on these comments, we propose moving either the data used for validation or the whole validation section to chapter 3. Furthermore, we propose specifying the corresponding validation results.

Page 17, line 10: "For instance, the event on the 21–22 June 2007 was caused by high intensity rainfall (Hilker et al., 2008). . ." Numerical data information desired, a range of the occurred rain intensities should be given (e.g. 5 mm h-1 over 30 hours, 200 mm d-1)

>The damaged objects caused by the thunderstorms on the 20–21 June 2007 are distributed over a large area including parts of the Western and Eastern Plateau as well as the Northern Alps. Therefore, the local rainfall varied greatly during this event. Nevertheless, to give a coarse indication of the rainfall intensities, we propose including the following information:

>For instance, the damages on the 20–21 June 2007 were caused by widespread thunderstorms with local rainfall intensities as high as 73 mmh-1 (Hilker et al., 2008) and are associated with a larger share of SWF damages (Fig. 11).

>While revising this passage, we realized that the corresponding dates are wrong. In Fig. 11 on page 19, the event is correctly labeled with 20–21 June 2007. In the text, however, the dates are erroneous and should be corrected, as specified above.

Page 20, Figure 12: dashed, dotted and solid ellipses are very hard to distinguish

>We propose increasing the gaps between the dots and dashes and/or increase the line width.

Page 22, Figure 14: red lines (Jura and Western Plateau) are very hard to distinguish too

>We thank the referee for pointing this out. Also, the second referee made a comment in this direction. We have experimented with different color and/or line patterns. However, we acknowledge that the choice might not be optimal, yet. Thus, we propose changing the color scheme for improving the figure's readability.

Chapter 5 – Discussion General remark: In this chapter the results of the presented project should be discussed more in context of supranational and international studies. The chapter lacks citations. So e.g. on Page 24, lines 26-30 and page 25, lines 1-4 (e.g. "Nevertheless, it is important to note that increasing absolute losses are most likely not attributable to climate change, but to socio- economic factors...")

>We thank the referee for this comment. The passage on page 24, lines 26-30, is solely based on the interpretation of our results (i.e. Fig. 13). Furthermore, the statement that damage in winter can most likely be attributed to local conditions such as frozen soils is based on a few case studies from various PICB. Thus, we do not think this passage needs further citations.

>The other mentioned passage, indeed, lacks the corresponding citations. We had erroneously omitted the corresponding references. We propose adding the citations as follows:

>Nevertheless, it is important to note that increasing absolute losses are most likely not attributable to climate change, but to socio-economic factors (e.g. Cutter and Emrich, 2005; Barredo, 2009; Bouwer, 2011; Kundzewicz et al., 2014).

Page 25, lines 5-24: Discussing methodological issuses. Should be put at the beginning of chapter 5 – Discussion

>We agree with the referee and propose moving the passage to the beginning of chapter 5 (Discussions).

General remarks of referee #2: Using a comprehensive data set with flood insurance claims from Switzerland representing 48% of all buildings in Switzerland, the authors have developed an algorithm that allows them to assign a flood type (fluvial flood or surface water flooding - SWF) to each damage claim using official flood hazard maps as further input. After that, they present a thorough analysis of the data revealing the relevance of surface water flooding, their regional as well as their temporal distribution. Throughout the analysis the number of claims and the total amount of loss are distinguished. By this, interesting insights in damaging processes and the differences between the two flood types are revealed. Overall, the paper addresses an important field and provides unique data and insights on the relevance of surface warter flooding, on which a lot of rumours, but only little evidence have existed so far. Despite this overall positive assessment, I would like to mention a few suggestions how to further improve the paper

>We greatly thank the referee for the time and effort to review our manuscript. We approve of most of the suggestions and propose changing the manuscript accordingly. The response to each specific point is listed in the following.

Specific comments of referee #2: 1) For an international audience the flood insurance scheme in Switzerland should be introduced in more detail.

>We think that a description of the Swiss insurance system beyond the mentioned main characteristics is not absolutely necessary. However, we realize that a proper citation is currently missing. Thus, we suggest including a reference to the review of the European natural hazard insurances by Schwarze et al. (2011). Therein, the Swiss system is described in detail. Moreover, the Swiss system is put into context with the system of other European countries. Thus, the interested reader may refer to this publication.

>Schwarze, R., Schwindt, M., Weck-Hannemann, H., Raschky, P., Zahn, F., and Wagner, G. G.: Natural hazard insurance in Europe: Tailored responses to climate change
are needed, Env. Pol. Gov., 21, 14–30, doi:10.1002/eet.554, 2011.

2) Section 2 should end with some recommendations on the usage of the terms/definitions or a clarification how they were used in the paper. In addition, the distinction between pluvial / SWF and fluvial floods should be explained.

>As the first referee raised similar points, we realize that particularly the end of chapter 2 should be improved. A possible solution could be the inclusion of a table summarizing the mentioned terms, as the first referee suggested. Moreover, the last paragraph could be moved or removed, if desired. Overall, we propose improving the corresponding passages.

>The distinction between pluvial floods and surface water floods is discussed on page 4, lines 27-33. Thus, we think that the distinction is sufficiently explained.

3) The usage of the data from the Swiss Mobilar is not clear to me. On p. 6, line 7, it is stated that those data were not used, but later they are mentioned several times. Please clarify. In addition, I would recommend to avoid the company's name throughout the paper to avoid (unintended) promotion.

>We used the data from the Swiss Mobiliar only for supporting the parametrization of the classification scheme. In order to clarify this, we propose changing the following two paragraphs. Moreover, as suggested by the referee, we agree that mentioning the company's name throughout the paper might lead to unintended promotion. Thus, we propose mentioning the company name only once and, thereafter, refer to the company only by the corresponding type of insurance company, e.g. by cooperative insurance company, abbreviated as CIC.

>In addition, we obtained similar records from the Swiss Mobiliar, a cooperative insurance company, hereafter referred to as CIC. Their data records were solely used to support the parametrization of the classification scheme. Thus, they were not part of the data analyses, as elaborated in more detail in Sect. 3.1.

In Sect. 3.1 (Data), we suggest altering the text as follows: The CIC's data contain flood damage claim records of content and, additionally, of property in cantons with no PICB. These records have quite similar characteristics as the data provided by the PICB, but are not limited to certain cantons and, thus, extend over the whole of Switzerland. However, unlike PICBs, the CIC does not hold a monopoly position. Consequently, the corresponding data records cover only the objects that are not insured by another private insurance company. Such records that are subjected to certain (unknown) market shares are much more challenging to interpret, as pointed out in the introduction. Nevertheless, the data are useful to set up the classification scheme, because every additional claim generally increases the method's robustness (c.f. Sect. 3.2). The data from the CIC are part of the data set D0, which is used solely to parametrize the classification scheme (c.f. Table 2 and Fig. 2).

4) The section 4.1 on validation comes a bit as a surprise. I would prefer to read the methodological consideratiions and the inforamtion about data available for validation in the section on Data and Methods. The results on the validation need more explanations.

>We thank the referee for raising this point. In fact, we had discussed where to place this section among the Co-Authors as well and decided to keep the whole section together. However, we agree that this is not the optimal solution. Following the referee's comment, we propose introducing the data we used for validation purposes in a new subsection "validation" in the chapter 3 (Material and methods). The results of the described validation would then be kept in the existing section 4.1. Alternatively, the whole section 4.1 could be moved to chapter 3, if desired.

5) Since the results are presented in several sub-sections, the reader has to wait a long time for a discussion and interpretation of the findings. Therefore, I would recommend to have a common section on Results and Discussion and add some discussion at the end of each subsection. In addition, more explanations and interpretations of the findings would be great, e.g. by considering the different nature of the flood processes

shown in Fig. 1, the spatial extent of triggering rainfall events and their seasonality. The advantages and shortcomings of the methodology could be shifted to the conclusions if this issue does not fit into the (new) more specific discussions.

>We acknowledge that there might be other forms to present and discuss our results. However, we find that in the current form the results and the discussions thereof are well structured. It is true that at the one hand this comes at the expense of having to wait longer for the corresponding interpretations. On the other hand, in this form we can discuss and interpret the results as a whole and not constrained to the topic of the respective sub-section. However, in accordance with a comment by the first referee, we propose restructuring chapter 5 (Discussions). Namely, we propose moving the discussion of methodical issues to the front of chapter 5. Thereby, we think the readability of the section is further improved and we hope that the form is then acceptable.

6) There are a lot of figures; some could be improved in quality and readability, particularly Fig. 13 and 14. In addition, the style of Fig. 11 is a bit confusing and full. Maybe you should concentrate on the 11 events only.

>We thank the referee for the feedback. Indeed, Figure 11 is rather full and might be unconventional. Although we had also considered to only showing the 11 highlighted events, we decided against it, because additional information would get lost. Specifically, it is shown that the bulk of all events cause relatively few damage claims. On the other hand, very few events cause substantially more claims, while coinciding with well-known major fluvial flood events. Finally, the connection of the highlighted events by lines shows that varying amounts of claims are registered at different days. That said, we find removing certain information from Figure 11 would compromise the figure's value. The challenge with Figure 13 and 14 is similar, as they both contain intersecting lines. As mentioned already in relation to a comment by the first referee concerning Figure 14, we have experimented with different color and/or line patterns. We found that by using colored and partially transparent lines we could produce the best results. However, we acknowledge that the choice might not be optimal, yet. Thus, we propose
changing the color scheme for improving the figures' readability.

7) The paper would also benefit from a language check.

>During revision, we will check the text once more and improve or correct passages and terms, as proposed in this document. We hope that the revised text will be acceptable and in line with the journal's standards.

Some minor points: - To my best knwoledge the plural of damage is not damages. Damages has a another meaning (in German: Schadensersatz/Entschädigung).

>We thank the referee for pointing this out. We were not aware of this subtlety and propose changing all corresponding occurrences of this term. Namely, we propose the general term "damage" to refer broadly to the number of damage claims as well as to the associated losses. When specifically referring to the occurred cases of damage, we propose the term "damage claims". Yet, when addressing the incurred losses, we propose the term "losses".

- p. 2, line 2: Besides Amsterdam, the city of Münster, Germany, experienced heavy pluvial flooding on the same day causing one of the costliest hazard events for German insurers (see Spekkers et al. 2017 for a comparison of the event in the two cities); see: http://www.nat-hazards-earth-syst-sci-discuss.net/nhess-2017-125/

>We thank the referee for this information and propose including the mentioned reference, as follows: At the same day in 2014 the Dutch capital Amsterdam (e.g., Gaitan et al., 2016; Spekkers et al., 2017) as well as Münster, Germany, experienced substantial flooding (Spekkers et al., 2017).

>Spekkers, M., Rözer, V., Thieken, A., ten Veldhuis, M.-c., and Kreibich, H.: A comparative survey of the impacts of extreme rainfall in two international case studies, Nat. Hazards Earth Syst. Sci. Discuss., pp. 1–38, doi:10.5194/nhess-2017-125, 2017.

- p 2: The role and relevance of market penetration of flood insurance should also be addressed here.

>The issue of market penetration is mentioned later, on page 3, line 13-18. We think that this issue is already appropriately placed and discussed.

- p 3: Insurance data generally do not include further information about factors and processes that explain the amount of damage as this information is usually not recorded by insurers to save costs. This should be addressed in this section.

>We have mentioned this issue in Sect. 3.2 on page 9, line 8-10. However, we agree that we can make that point earlier, as well. Therefore, we propose integrating that information at an earlier stage, as well.

- p. 3, line 33: The term "direct tangible flood losses" should be briefly defined. Add some information (maybe in the section on methods) on how you treated deductibles and upper bounds of insurance coverage (if applicable).

>We propose adding the required short definition adapted to the case of direct tangible flood damage to buildings based on the comprehensive review by Merz et al. (2010). We propose altering the passage as follows:

>As the PICB, safe a few exceptions, insure only property and not its contents, this study only considers damages to buildings. More specifically, this study is limited to direct tangible flood damages to buildings, i.e. monetary losses caused by the buildings' direct contact with flood water (Merz et al., 2010). Thereby, we acknowledge that these damages only constitute part of the total flood losses.

>Merz, B., Kreibich, H., Schwarze, R., and Thieken, A.: Review article "Assessment of economic flood damage", Nat. Hazard Earth Sys., 10, 1697–1724, doi:10.5194/nhess-10-1697-2010, 2010.

>We also propose adding some information about deductibles and upper bounds of insurance coverage (not applicable). Thus, we suggest extending the corresponding passage as follows:

>In terms of loss, we assessed total loss values, i.e. the sum of the registered pay offs

and applicable deductibles. Since the insurance coverage is not limited to an upper bound, the maximal total loss for each building equals its sum insured. Applicable deductibles vary between the different PICBs, whereas no deductibles at all, a fixed participation of a few hundred Swiss francs or a variable participation of 10 % within a fixed range with a maximum value of CHF~4000 are applied.

- p.6, line 17-23: Fig 2 and Table 1 should be shifted to the section in which both are explained in detail.

>We apologize for the fact that some figures and all tables are not located within the sections they belong. However, the manuscript was produced with the journal's LaTeX template, with which the user has limited control over where exactly the figures are placed. Concerning the tables, they were all placed at the end of the manuscript, as required by the journal's guidelines. However, if the manuscript is accepted for publication, this issue will (hopefully) be taken care of in the final layout.

- p.8, line 7: Could you provide examples for data errors and their corrections?

>In accordance with a request for a more specific statement by the first referee and a request for examples by the second referee, we propose changing this passage as follows: During this procedure, the data were quality checked. Obvious errors such as address misspellings or flipped coordinate pairs were corrected. Furthermore, we removed duplicated entries, as well as records with incomplete (e.g. missing address) or invalid data (e.g. invalid damage date).

- p.8, line 8: To which index is the used index comparable? What are the underlying counted goods and services?

>The indices applied by the PICB are commonly based on construction output prize indices. Thus, we propose including this information in the corresponding passage.

- p.9, line 22-23: This process is not 100% clear.

>We suggest elaborating this passage in more detail, as follows:

>Before calculating the Euclidean distance to the next river, we first mask the parts of the river network that are located at lower altitudes than the respective object. For that matter, we create a raster mask indicating cells that are located at the same or at higher altitudes than the corresponding object based on a digital elevation model (DEM, Table 1). Only then, the Euclidean distance to the masked river network is assessed.

- p.10, line 1-2: In my view, this (lake inundations are covered by fluvial flood haazrd maps) is the most convincing point for the assumption made and should be mentioned earlier.

>We realized that the formulation might not be clear enough. Namely, we have to differentiate the influence of damage caused by overflowing lakes on the parametrization of the classification scheme, from the influence of the application of this scheme on the classification itself. Firstly, we elaborate the effect of claims associated with overflowing lakes on the classification scheme's parametrization. Secondly, we elaborate to which extent this is influencing the result, in case we apply this classification scheme. To clarify this, we suggest altering the corresponding paragraph as follows:

>Inherently, the flood claims also include damage caused by overflowing lakes, which could not be distinguished easily from fluvial floods. Consequently, damage related to lakes will be associated with a certain distance to the next river, even though the corresponding river was not the cause of the damage. A visual check of such claims revealed that they tend to be located closer to the corresponding lake than the next watercourse. Technically, this shifts the ECDF of distances to the right and, accordingly, renders higher percentile values (c.f. Fig. 4 and Table 3). In turn, applying the classification scheme with increased percentile values, leads to more claims being associated with fluvial floods instead of SWFs. However, as the number of claims associated with overflowing lakes is low in comparison to claims associated with overtopping rivers, it is safe to assume that this influence is negligible. At most, it might lead to a slightly more conservative classification of SWF claims. Besides, the claims associated with overflowing lakes are directly and correctly classified as fluvial floods, because the hazard

[Figure]

of overflowing lakes is consistently considered in the fluvial flood maps (c.f. Fig. 5).

- p.10, line 15: check terms (use hazard instead of danger)

>We agree that the term "danger" is not quite appropriate. However, we adopted the terms used in the cited document that differentiates hazard map (general term) and danger maps (specific product with associated so-called "danger" levels). To clarify this, we have suggest changing the passage as follows:

>Within these perimeters, the fluvial flood hazards are indicated using four different so-called danger levels (de Moel et al., 2009), whereas we define the flood hazard zone as the combined area of low, medium and elevated danger, while excluding the area categorized as residual danger (c.f. Zimmermann et al., 2005).

- p.12, line 20: Is there any evidence for this assumption?

>This assumption is based on the fact that SWFs have not been considered by any building code, so far. Moreover, the considered period is several times shorter than a regular life span of Swiss buildings. Thus, we suggest integrating this information in the corresponding passage.

- p.15, line 22-23: Please add: how did you deal with consecutive days with claims when defining events?

To elaborate this issue further, we suggest expanding the passage as follows:

>For that matter, we have stratified the data according to the total number of claims per day using five categories ranging from single (1-5 claims per day) to vast (>501 claims per day). We defined an event as a day with at least one claim of any class (A-E), which amounts to a total of 1'490 events in the period of 1999-2013. Obviously, this is a pragmatic definition of an event. Specifically, separate local events occurring at the same day are counted as a single event, while events spanning over several days are counted as individual events. Nevertheless, the pragmatic definition is sufficient for the purpose of a first simple analysis, presented hereafter.

- p.18, line 6: damage density should be defined (and refer to the appendix explicitly)

>As we have only used this term in this paragraph, we suggest removing this term altogether and use the term "relative values" which we can define in the given context as the number of damage claims in relation to the number of buildings within the same raster cell.

- p.26, line 8-9: The same is true for Germany, but the evidence is lacking. Maybe this point should be made clearer - already in the introduction.

>We thank the referee for this interesting addition. As this statement holds true for at least Switzerland and Germany, we think it is safe to make a more general statement by omitting the explicit focus on Switzerland. Thus, we suggest altering the passage as follows:

>There seems to be a consensus among practitioners and experts that SWFs are responsible for a large share of all flood damage. However, this perception does not stem from quantitative research, but rather from single case studies or practical experience and, thus, lacks evidence.

- p.27, line 23: Why? Please mention your assessment criteria.

>We chose a balance between level of detail and smoothness based on visual comparison of the produced output maps. The finer the resolution, the patchier the picture gets, while coarser resolution spatially smooth the results. We propose specifying this passage.

I am looking forward to the revised paper. Thank you for this interesting analysis.

Please also note the supplement to this comment:
https://www.nat-hazards-earth-syst-sci-discuss.net/nhess-2017-136/nhess-2017-136-AC1-supplement.pdf

---

## Author Response (AR1)

**"Surface water floods in Switzerland: what insurance claim records tell us about the caused damage in space and time" by Daniel B. Bernet, Volker Prasuhn, and Rolf Weingartner**

*Coments to the revised manuscript*

Once again, we would like to warmly thank the two anonymous referees as well as the editors for their time and effort to review this manuscript. We greatly appreciate their valuable and constructive inputs that helped to improve the manuscript's quality.

In the following, we go through each of the raised points by the two referees and indicate the corresponding changes. The changes are highlighted in the revised manuscript appended to the end of this document, whereas blue colored passages refer to the additions and crossed out passages in red refer to removed sections. The page and line numbers refer to the revised manuscript.

**General remarks of referee #1:**

1. The paper discusses a very interesting theme as surface water floods are felt as an increasing problem in intensively settled areas more and more and demand for research on this theme is increasing. Thus, the paper gives significant contribution on objectifying the discussion on losses due to natural hazards like surface water flow (SWF) and matches the scope of NHESS.
2. In addition the paper presents a new way to interpret insurance data in terms of natural hazards.
3. Tools, methods, results are up to international standards.
4. Results of the investigations strongly support the interpretation and the conclusions drawn in the paper.
5. Desicription of data and methods used is sufficient except three points (pages 3, 9 and 12, see further comments below).
6. The title clearly describes the main concern of the paper
7. Title and abstract are clearly presented and easy to read
8. Connex to previous and related research is sufficently described in chapter 1
9. Number of references should be enlarged in chapter 5. The results of the presented project should be discussed more in context of supranational and international studies (see further comments below)
10. Presentation of the paper is structured well and clearly presented except parts of chapters 2 and five (see further comments below)
11. Length of the paper seems adequate
12. Figures and tables are clearly presented, except figures 12 and 14 (some symbols are hard to distinguish)
13. Technical and english language are of good quality, fluently presented and understandable for an interdisciplinary audience

> *We greatly thank the referee for the time and effort to review our manuscript. We approve of most of the suggestions and changed the manuscript accordingly. The response to each specific point is listed in the following.*

**Specific comments of referee #1:**

**Ad chapter Introduction**

Gives a good overview about the actual situation.

**Page 2, line 16-17:** „In order to reduce the risk, it is suggested to focus on the physical protection of exposed objects (e.g., Kron, 2009; DWA, 2013)…."

Remark: This is one point of concern. But one step before: Adaptation and improvement of spatial management (hazard zone mapping, spatial development / management plans etc.) to reduce construction activities or to allow building activities only with special obligations in areas prone to SWF is necessary.

> *We thank the referee for the valuable remark. We agree that the hazard of SWF should definitely be taken into account in spatial planning and should be managed accordingly. However, a prerequisite is that the hazardous process is understood well and can be predicted. In the cited documents, the authors argue that surface water floods (SWFs) can occur practically everywhere and are difficult to predict. Thus, they argue that the most effective measure is to focus on the physical protection of the object itself. In order to clarify that statement, we changed the citation as follows:*

> **Page 2, line 19 f.:** *In order to reduce the risk, an effective approach is to focus on the physical protection of exposed objects (e.g., Kron, 2009; DWA, 2013).*

**Ad chapter 2 – Terminology**

Presented in a more descriptive manner. From the announcement in the last paragraph of chapter 1 (page 3) a short and precise presentation of the most important terms, a precise definition for every term has been expected. Thus, presentation in form of a table with precise definionitions would be useful, in addition the chapter could be shortened and kept more precise.

**Page 5, Figure 1:** A title / precise description is missing – what is Fig.1 showing?

> *Also considering the comment by the second reviewer, we rewrote the whole chapter. We tried to improve the text in accordance to the comments of the two reviewers. In addition, we introduced a table concisely summarizing the mentioned terms (c.f. **Page 39, Table 1**). Moreover, we added a precise title of Fig. 1 and better integrated the Figure's content. Please refer to the revised manuscript, which visualizes the corresponding changes.*

**Page 5, lines 15-17, page 6, lines 1-5:** No contribution of this text to improve systematic understanding. When it is the authors' opinion that these explanations are absolutely necessary – put them at the beginning of this chapter.

> *We agree that the paragraph does not improve the systematic understanding of different terms. However, the paragraph was meant to give some possible explanation and background information about why the classification of flood processes is not simple. Therefore we moved the passage to the beginning of chapter 2 and altered it accordingly. Please refer to the revised manuscript, which visualizes the corresponding changes.*

**Ad Chapter 3 – Material and methods**

**Page 6, line 31:** „The provided data provided by…" – improvement of wording necessary

> *We thank the referee for pointing this out. However, as we changed this sentence following a comment of the second referee, this correction became obsolete.*

**Page 8, line 6-7:** „…obvious errors were corrected, if possible, or removed otherwise…". Suggestion for further specification: „…obvious errors were corrected, if possible, or the invalid data removed otherwise…"

> *We acknowledge that this passage is not specific enough. In accordance with the second referee's request for some examples, we changed this passage as follows:*

*Page 9, line 19 ff.: During this procedure, the data were quality checked. Obvious errors such as address misspellings or flipped coordinate pairs were corrected. Furthermore, we removed duplicated entries, as well as records with incomplete (e.g. missing address) or invalid data (e.g. invalid damage date).*

**Page 9, line 16:** „…Thereby, the question ist how these distance…"

Suggestion: „…Thereby, the question is how this distance…"

*We thank the referee for pointing out the error and changed it as suggested.*

**Page 9, lines 30-31:** „…To take regional geographical characteristics into account, the percentiles are calculated for each region separately (Table 3)…".

Question: How have special geographical characteristics been taken into account for the different regions. Give a short description or a reference where the procedure has been described alrady.

*We agree with the referee that this statement is not quite clear. The regions, by which the data are grouped and stratified, represent areas with similar geographical characteristics. Specifically, the regions are based on a classification of Switzerland into natural landscape units after Grosjean (1975). At the coarsest level of the presented classification, the author considered mainly orographic as well as climatic factors. As mentioned on page 6, lines 26-28, we adapted these regions, while constraining the regions' borders to the boundaries of hydrological catchments. Thus, by assessing the percentiles separately for each region, we implicitly take the geographical characteristics, by which the corresponding regions were classified, into account. To clarify the passage, we propose changing the text as follows:*

*Page 11, line 21 ff.: The percentiles are calculated for each region separately (Table 4). As the regions themselves represent areas with similar orographic and climatic characteristics (Grosjean, 1975), we thereby implicitly take these regional geographical characteristics into account.*

*Furthermore, we realized that on **page 8, line 3**, instead of "…adopted…" it should read "…adapted…" We changed this accordingly.*

**Page 11, Figure 5:** wrong: „…Intersects Aqua-portect flood zone…"

Suggestion: „…Intersects Aqua-protect flood zone…"

*We changed the misspelling in Figure 5 according to the referee's comment.*

**Page 11, lines 4-5:** „…The smallest rivers were not considered by Aquaprotect, but there is no objective way of knowing where the cut-off was set…"

Question: Definition of the term „the smallest rivers". A more precise information is necessary. Brooklets with l/sec, brooks with a few m³/sec – which type of running water?

*We agree that the expression "smallest river" is not appropriate. However, as mentioned in the corresponding passage, we could not find any information concerning a cut-off criterion or a method to obtain this information indirectly. The Aquaprotect's documentation does not provide any indication concerning this issue. Thus, it remains unclear whether a minimum flow threshold, a minimum flow accumulation or catchment area or a similar criterion was considered. It is apparent, however, that Aquaprotect consistently does not cover headwaters. Similarly, small tributaries are not considered. However, it is not possible to infer and report a quantitative threshold from this data. Nevertheless, we changed the passage as follows:*

*Page 13, line 3 f.: Consistently, headwaters and small tributaries are not covered by Aquaprotect. Yet, no information about the specific exclusion criterion could neither be found nor inferred.*

**Page 12, lines2-4:** „…Nevertheless, the level of detail of the Swiss flood hazard maps far exceeds the one of Aquaprotect. This is taken into account by applying lower percentile levels for claims located within the flood hazard perimeters, as shown in Fig. 5 and 6…"

Question: On which basis of the lower percentile levels have been assessed? Systematic algorithm / empirical decision?

*We chose the lower percentile levels empirically, based on case studies. To reflect this decision, we changed the statement as follows:*

*__Page 14, line 3 f.:__ We considered this by empirically choosing lower percentile levels for claims located within the flood hazard perimeters, as shown in Fig. 5 and 6.*

**Ad chapter 4 – Results**

**Page 13, lines 4-8:** The passus „In Switzerland, there are few quantitative studies…..and covers only a small part of Switzerland." should be moved to chapter 5 – Discussion.

*This passage was meant as an introduction to chapter 4 (Results), whereby the results to be presented are put into context. However, we agree that it belongs to chapter 5 (Discussions). However, as we had to integrate it smoothly, we had to alter the passage accordingly. The passage can now be found in chapter 5 on __Page 26, line 11 ff.__ in the revised manuscript.*

**Page 13, lines 30-31:** „Nevertheless, the indicated hazard zones were mostly located in the vicinity of SWF claims."

Improved precision of the statement necessary. Suggestion: „Nevertheless, the indicated hazard zones were mostly located in the vicinity of SWF claims, forming the data-basis of the investigations presented in this paper"

*We thank the referee for the suggested improvement. Also, the second referee requested a specification of this passage (see corresponding comment further below). The second referee proposed to move the whole chapter or at least parts thereof to chapter 3 (Materials and methods). Based on these comments, we moved the data used for validation to chapter 3 and kept the application of the validation in chapter 4. Furthermore, we specified the corresponding validation results, also specifying the passage mentioned by the first referee. The improved section 3.1 starts on __Page 15, line 19__, while section 4.1 starts on __Page 16, line 22__. Please refer to the revised manuscript, which visualizes the corresponding changes.*

**Page 17, line 10:** „For instance, the event on the 21–22 June 2007 was caused by high intensity rainfall (Hilker et al., 2008)…"

Numerical data information desired, a range of the occurred rain intensities should be given (e.g. 5 mm h$^{-1}$ over 30 hours, 200 mm d$^{-1}$)

*The damaged objects caused by the thunderstorms on the 20–21 June 2007 are distributed over a large area including parts of the Western and Eastern Plateau as well as the Northern Alps. Therefore, the local rainfall varied greatly during this event. Nevertheless, to give a coarse indication of the rainfall intensities, we included the following information:*

*__Page 22, line 7 ff.__: For instance, the damage on the 20–21 June 2007 was caused by widespread thunderstorms with local rainfall intensities as high as 73 mmh$^{-1}$ (Hilker et al., 2008) and is associated with a larger share of SWF damage claims (Fig. 11).*

*While revising this passage, we realized that the corresponding dates are wrong. In Fig. 11 on page 21, the event is correctly labeled with 20–21 June 2007. In the text, however, the dates are erroneous and were corrected, as specified above.*

**Page 20, Figure 12:** dashed, dotted and solid ellipses are very hard to distinguish

*We increased the line width for increased readability according to the referee's comment.*

**Page 22, Figure 14:** red lines (Jura and Western Plateau) are very hard to distinguish too

*We thank the referee for pointing this out. Also, the second referee made a comment in this direction. Indeed the red/orange lines were difficult to distinguish. Therefore, we have experimented with different color patterns. We could improve the readability by reducing the transparency. Moreover, we adjusted the colors to create better contrasts between the lines. Nevertheless, overlapping lines cannot be avoided. Nevertheless, we think the figure gets the discussed points across, in particular that the peak values of most regions are in phase with each other. The improved Figure 14 can be found on **Page 25** in the revised manuscript.*

**Chapter 5 – Discussion**

**General remark:** In this chapter the results of the presented project should be discussed more in context of supranational and international studies. The chapter lacks citations. So e.g. on

**Page 24, lines 26-30 and page 25, lines 1-4** (e.g. „Nevertheless, it is important to note that increasing absolute losses are most likely not attributable to climate change, but to socio- economic factors…")

*We thank the referee for this comment. The passage on page 24, lines 26-30, is solely based on the interpretation of our results (i.e. Fig. 13). Furthermore, the statement that damage in winter can most likely be attributed to local conditions such as frozen soils is based on a few case studies from various PICB. Thus, we do not think this passage needs further citations.*

*The other mentioned passage, indeed, lacks the corresponding citations. We had erroneously omitted the corresponding references. We added the citations as follows:*

***Page 28, line 34 ff.**: Nevertheless, it is important to note that increasing absolute losses are most likely not attributable to climate change, but to socio-economic factors (e.g. Cutter and Emrich, 2005; Barredo, 2009; Bouwer, 2011; Kundzewicz et al., 2014).*

**Page 25, lines 5-24:** Discussing methodological issues. Should be put at the beginning of chapter 5 – Discussion

*We agree with the referee and moved the passage to the beginning of chapter 5 (Discussions). We had to slightly adjust the passage to guarantee smooth transitions. The corresponding passage is now starting on **Page 26, line 11**. Please refer to the revised manuscript, which visualizes the corresponding changes.*

**General remarks of referee #2:**

Using a comprehensive data set with flood insurance claims from Switzerland representing 48% of all buildings in Switzerland, the authors have developed an algorithm that allows them to assign a flood type (fluvial flood or surface water flooding - SWF) to each damage claim using official flood hazard maps as further input. After that, they present a thorough analysis of the data revealing the relevance of surface water flooding, their regional as well as their temporal distribution. Throughout the analysis the number of claims and the total amount of loss are distinguished. By this, interesting insights in damaging processes and the differences between the two flood types are revealed. Overall, the paper addresses an important field and provides unique data and insights on the relevance of surface warter flooding, on which a lot of rumours, but only little evidence have existed so far.

Despite this overall positive assessment, I would like to mention a few suggestions how to further

improve the paper

*We greatly thank the referee for the time and effort to review our manuscript. We approve of most of the suggestions and propose changing the manuscript accordingly. The response to each specific point is listed in the following.*

**Specific comments of referee #2:**

1) For an international audience the flood insurance scheme in Switzerland should be introduced in more detail.

*We think that a description of the Swiss insurance system beyond the mentioned main characteristics is not absolutely necessary. However, we realize that a proper citation is currently missing. Thus, we included a reference to the review of the European natural hazard insurances by Schwarze et al. (2011). Therein, the Swiss system is described in detail. Moreover, the Swiss system is put into context with the system of other European countries. Thus, the interested reader may refer to this publication. The reference is located on **Page 3, line 26.***

*The corresponding reference was added to the reference list. The new reference can be found on **Page 37, line 17 f.***

*Schwarze, R., Schwindt, M., Weck-Hannemann, H., Raschky, P., Zahn, F., and Wagner, G. G.: Natural hazard insurance in Europe: Tailored responses to climate change are needed, Env. Pol. Gov., 21, 14–30, doi:10.1002/eet.554, 2011.*

2) Section 2 should end with some recommendations on the usage of the terms/definitions or a clarification how they were used in the paper. In addition, the distinction between pluvial / SWF and fluvial floods should be explained.

*As the first referee raised similar points, we realize that particularly chapter 2 needed improvements. We have revised and rewritten the whole chapter. We included a table summarizing the mentioned terms, as the first referee suggested. Moreover, the last paragraph was moved to the beginning of the chapter. The revised chapter starts on **Page 4, line 12**. Please refer to the revised manuscript, which visualizes the corresponding changes.*

*Note that the distinction between pluvial floods and surface water floods has been improved and is now discussed on P**age 6, lines 1 ff.***

3) The usage of the data from the Swiss Mobilar is not clear to me. On p. 6, line 7, it is stated that those data were not used, but later they are mentioned several times. Please clarify. In addition, I would recommend to avoid the company's name throughout the paper to avoid (unintended) promotion.

*We used the data from the Swiss Mobiliar only for supporting the parametrization of the classification scheme. In order to clarify this, we propose changing the following two paragraphs. Moreover, as suggested by the referee, we agree that mentioning the company's name throughout the paper might lead to unintended promotion. Thus, we mentioned the company name only once and, thereafter, refer to the company only by the corresponding type of insurance company, e.g. by cooperative insurance company, abbreviated as CIC. Please refer to the revised manuscript, which visualizes all occurrences of this substitution. Moreover, the following sections were changed as follows:*

***Page 7, line 11 ff.**: In addition, we obtained similar records from the Swiss Mobiliar, a cooperative insurance company, hereafter referred to as CIC. The corresponding data records were solely used to support the parametrization of the classification scheme. Thus, they were not part of the data analyses, as elaborated in more detail in Sect. 3.1.*

*In Sect. 3.1 (Data), we altered the text as follows:*

*Page 9, line 7 ff.: The CIC's data contain flood damage claim records of content and, additionally, of property in cantons with no PICB. These records have quite similar characteristics as the data provided by the PICB, but are not limited to certain cantons and, thus, extend over the whole of Switzerland. However, unlike PICBs, the CIC does not hold a monopoly position. Consequently, the corresponding data records cover only the objects that are not insured by another private insurance company. Such records that are subjected to certain (unknown) market shares are much more challenging to interpret, as pointed out in the introduction. Nevertheless, the data are useful to set up the classification scheme, because every additional claim generally increases the method's robustness (c.f. Sect. 3.2). The data from the CIC are part of the data set D0, which is used solely for parametrizing the classification scheme (c.f. Table 3 and Fig. 2).*

4) The section 4.1 on validation comes a bit as a surprise. I would prefer to read the methodological consideratiions and the inforamtion about data available for validation in the section on Data and Methods. The results on the validation need more explanations.

*We thank the referee for raising this point. In fact, we had discussed where to place this section among the Co-Authors as well and decided to keep the whole section together. However, we agree that this is not the optimal solution. Following the referee's comment, we introduce the data we used for validation purposes in a new subsection "validation" in the chapter 3 (Material and methods). The results of the described validation are kept in the existing section 4.1. The improved section 3.1 starts on **Page 15, line 19**, while section 4.1 starts on **Page 16, line 22**. Please refer to the revised manuscript, which visualizes the corresponding changes.*

5) Since the results are presented in several sub-sections, the reader has to wait a long time for a discussion and interpretation of the findings. Therefore, I would recommend to have a common section on Results and Discussion and add some discussion at the end of each subsection. In addition, more explanations and interpretations of the findings would be great, e.g. by considering the different nature of the flood processes shown in Fig. 1, the spatial extent of triggering rainfall events and their seasonality. The advantages and shortcomings of the methodology could be shifted to the conclusions if this issue does not fit into the (new) more specific discussions.

*We acknowledge that there might be other forms to present and discuss our results. However, we find that in the current form the results and the discussions thereof are well structured. It is true that at the one hand this comes at the expense of having to wait longer for the corresponding interpretations. On the other hand, in this form we can discuss and interpret the results as a whole and not constrained to the topic of the respective sub-section. However, in accordance with a comment by the first referee, we restructured chapter 5 (Discussions). Namely, we moved the discussion of methodical issues to the front of chapter 5. Thereby, we think the readability of the section has further improved and we hope that the form is now acceptable.*

6) There are a lot of figures; some could be improved in quality and readability, particularly Fig. 13 and 14. In addition, the style of Fig. 11 is a bit confusing and full. Maybe you should concentrate on the 11 events only.

*We thank the referee for the feedback. Indeed, Figure 11 is rather full and might be unconventional. Although we had also considered to only showing the 11 highlighted events, we decided against it, because additional information would get lost. Specifically, it is shown that the bulk of all events cause relatively few damage claims. On the other hand, very few events cause substantially more claims, while coinciding with well-known major fluvial flood events. Finally, the connection of the highlighted events by lines shows that varying amounts of claims are registered at different days. That said, we find removing certain information from Figure 11 would compromise the figure's value and thus kept the figure as it was.*

*The challenge with Figure 13 and 14 is similar, as they both contain intersecting lines. As*

*mentioned already in relation to a comment by the first referee concerning Figure 14, we agree that the particularly red/orange lines were difficult to distinguish. Therefore, we have experimented with different color patterns. We could improve the readability by reducing the transparency. Moreover, we adjusted the colors to create better contrasts between the lines. Nevertheless, overlapping lines cannot be avoided. However, we find that the main patterns are displayed well enough and support the findings and discussions in this study. The improved Figure 13 can be found on **Page 24** and Figure 14 on **Page 25** of the revised manuscript.*

*Following the comment of the second referee, we further improved Figure 6. Namely, the pattern of the Aquaprotect flood zone was not well displayed. Thus, we changed the pattern to a dark shade of gray. The improved figure can be found on **Page 14** of the revised manuscript.*

7) The paper would also benefit from a language check.

*During revision, we checked the text once more and improved or corrected passages, as outlined in this document. Moreover, we changed certain terms that had been used erroneously. For instance, we substituted "methodology" by "methods" and revised the term "thereby", which was not used correctly in all occurrences. Please refer to the revised manuscript, which visualizes all occurrences of these substitutions.*

Some minor points:

- To my best knwoledge the plural of damage is not damages. Damages has a another meaning (in German: Schadensersatz/Entschädigung).

*We thank the referee for pointing this out. We were not aware of this subtlety and changed all corresponding occurrences of this term. Namely, we propose the general term "damage" to refer broadly to the number of damage claims as well as to the associated losses. When specifically referring to the occurred cases of damage, we propose the term "damage claims". Yet, when addressing the incurred losses, we propose the term "losses". Please refer to the revised manuscript, which visualizes all occurrences of this substitution.*

- p. 2, line 2: Besides Amsterdam, the city of Münster, Germany, experienced heavy pluvial flooding on the same day causing one of the costliest hazard events for German insurers (see Spekkers et al. 2017 for a comparison of the event in the two cities); see: http://www.nat-hazards-earth-syst-sci-discuss.net/nhess-2017-125/

*We thank the referee for this information and included the mentioned reference, as follows:*

*__Page 2, line 4 ff.__: At the same day in 2014, the Dutch capital Amsterdam (e.g., Gaitan et al., 2016; Spekkers et al., 2017) as well as Münster, Germany, experienced substantial flooding (Spekkers et al., 2017).*

*The corresponding reference was added to the reference list. The new reference can be found on **Page 37, line 19 f.***

*Spekkers, M., Rözer, V., Thieken, A., ten Veldhuis, M.-c., and Kreibich, H.: A comparative survey of the impacts of extreme rainfall in two international case studies, Nat. Hazards Earth Syst. Sci. Discuss., pp. 1–38, doi:10.5194/nhess-2017-125, 2017.*

- p 2: The role and relevance of market penetration of flood insurance should also be addressed here.

*The issue of market penetration is mentioned later, on **Page 3, line 16-22**. We think that this issue is already appropriately placed and discussed.*

- p 3: Insurance data generally do not include further information about factors and processes that explain the amount of damage as this information is usually not recorded by insurers to save costs. This should be addressed in this section.

*We have mentioned this issue in Sect. 3.2 on **Page 10, line 6-8**. However, we agree that we can make that point earlier, as well. Therefore, we integrated that information at an earlier stage,*

*as well.*

**Page 3, line 10 ff.**: *As insurance companies usually do not assess and record detailed information for each damage claim, it is difficult to verify and differentiate the cause of each damage without at least knowing the explicit location of the damaged object.*

- p. 3, line 33: The term "direct tangible flood losses" should be briefly defined. Add some information (maybe in the section on methods) on how you treated deductibles and upper bounds of insurance coverage (if applicable).

*We added the required short definition adapted to the case of direct tangible flood damage to buildings based on the comprehensive review by Merz et al. (2010). We altered the passage as follows:*

**Page 4, line 1 ff.**: *As the PICB, safe a few exceptions, insure only property and not its contents, this study only considers damages to buildings. More specifically, this study is limited to direct tangible flood damages to buildings, i.e. monetary losses caused by the buildings' direct contact with flood water (Merz et al., 2010). Thereby, we acknowledge that these damages only constitute part of the total flood losses.*

*The corresponding reference was added to the reference list. The new reference can be found on* **Page 36, line 35 f.**

*Merz, B., Kreibich, H., Schwarze, R., and Thieken, A.: Review article "Assessment of economic flood damage", Nat. Hazard Earth Sys., 10, 1697–1724, doi:10.5194/nhess-10-1697-2010, 2010.*

*We also added some information about deductibles and upper bounds of insurance coverage (not applicable). Thus, we extended the corresponding passage as follows:*

**Page 9, line 22 ff.**: *In terms of loss, we assessed total loss values, i.e. the sum of the registered pay offs and applicable deductibles. Since the insurance coverage is not limited to an upper bound, the maximal total loss for each building equals its sum insured. Applicable deductibles vary between the different PICBs, whereas no deductibles at all, a fixed participation of a few hundred Swiss francs or a variable participation of 10 % within a fixed range with a maximum value of CHF~4000 are applied.*

- p.6, line 17-23: Fig 2 and Table 1 should be shifted to the section in which both are explained in detail.

*We apologize for the fact that some figures and all tables are not located within the sections they belong. However, the manuscript was produced with the journal's LaTeX template, with which the user has limited control over where exactly the figures are placed. Concerning the tables, they were all placed at the end of the manuscript, as required by the journal's guidelines. However, if the manuscript is accepted for publication, this issue will (hopefully) be taken care of in the final layout.*

- p.8, line 7: Could you provide examples for data errors and their corrections?

*In accordance with a request for a more specific statement by the first referee and a request for examples by the second referee, we propose changing this passage as follows:*

**Page 9, line 19 ff.**: *During this procedure, the data were quality checked. Obvious errors such as address misspellings or flipped coordinate pairs were corrected. Furthermore, we removed duplicated entries, as well as records with incomplete (e.g. missing address) or invalid data (e.g. invalid damage date).*

- p.8, line 8: To which index is the used index comparable? What are the underlying counted goods and services?

*The indices applied by the PICB are commonly based on construction output prize indices. Thus,*

*we included this information in the corresponding passage, as follows:*

*Page 9, line 26 ff.: Finally, the total loss values were corrected for inflation as of 2013 by applying the respective construction output price index considered by each PICB, in case the source data had not been indexed already.*

- p.9, line 22-23: This process is not 100% clear.

*We elaborated this passage in more detail, as follows:*

*Page 11, line 9 ff.: Before calculating the Euclidean distance to the next river, we first mask the parts of the river network that are located at lower altitudes than the respective object. For this task, we create a raster mask indicating cells that are located at lower altitudes than the corresponding object, based on a digital elevation model (DEM, Table 2). The Euclidean distance to the river network is then assessed by using the raster mask, which hides all river sections at lower altitudes than the respective object.*

- p.10, line 1-2: In my view, this (lake inundations are covered by fluvial flood haazrd maps) is the most convincing point for the assumption made and should be mentioned earlier.

*We realized that the formulation might not be clear enough. Namely, we have to differentiate the influence of damage caused by overflowing lakes on the parametrization of the classification scheme, from the influence of the application of this scheme on the classification itself. Firstly, we elaborate the effect of claims associated with overflowing lakes on the classification scheme's parametrization. Secondly, we elaborate to which extent this is influencing the result, in case we apply this classification scheme. To clarify this, we altered the corresponding paragraph as follows:*

*Page 11, line 24 ff.: Inherently, the flood claims also include damage caused by overflowing lakes, which could not be distinguished easily from fluvial floods. Consequently, damage related to lakes will be associated with a certain distance to the next river, even though the corresponding river was not the cause of the damage. A visual check of such claims revealed that they tend to be located closer to the corresponding lake than the next watercourse. Technically, this shifts the ECDF of distances to the right and, accordingly, renders higher percentile values (c.f. Fig. 4 and Table 4). In turn, applying the classification scheme with increased percentile values, leads to more claims being associated with fluvial floods instead of SWFs. However, as the number of claims associated with overflowing lakes is low in comparison to claims associated with overtopping rivers, it is safe to assume that this influence is negligible. At most, it might lead to a slightly more conservative classification of SWF claims. Besides, the claims associated with overflowing lakes are directly and correctly classified as fluvial floods, because the hazard of overflowing lakes is consistently considered in the fluvial flood maps (c.f. Fig. 5).*

- p.10, line 15: check terms (use hazard instead of danger)

*We agree that the term "danger" is not quite appropriate. However, we adopted the terms used in the cited document that differentiates hazard map (general term) and danger maps (specific product with associated so-called "danger" levels). To clarify this, we changed the passage as follows:*

*Page 12, line 10 ff.: Within these perimeters, the fluvial flood hazards are indicated using four different so-called danger levels (de Moel et al., 2009), whereas we define the flood hazard zone as the combined area of low, medium and elevated danger, while excluding the area categorized as residual danger (c.f. Zimmermann et al., 2005).*

- p.12, line 20: Is there any evidence for this assumption?

*This assumption is based on the fact that SWFs have not been considered by any building code, so far. Moreover, the considered period is several times shorter than a regular life span of Swiss*

*buildings. Thus, we integrated this information in the corresponding passage, as follows:*

*__Page 15, line 1 ff.__: This assumption seems appropriate since SWFs have not been considered by any building code, so far. Moreover, the analyzed period is several times shorter than the regular life span of Swiss buildings. Lastly, we apply the seasonal Mann-Kendall test (Hirsch et al., 1982) with a significance level of 0.1 for the resulting p-value to test whether the number of damage claims and associated losses have increased or decreased over time.*

- p.15, line 22-23: Please add: how did you deal with consecutive days with claims when defining events?

*To elaborate this issue further, we expanded the passage as follows:*

*__Page 20, line 1 ff.__: Obviously, this is a pragmatic definition of an event. Specifically, separate local events occurring at the same day are counted as a single event, while events spanning over several days are counted as individual events. Nevertheless, the pragmatic definition is sufficient for the purpose of a first simple analysis, presented hereafter.*

- p.18, line 6: damage density should be defined (and refer to the appendix explicitly)

*As we have only used this term in this paragraph, we removed this term altogether and used the term "relative values" which we can define in the given context as the number of damage claims in relation to the number of buildings within the same raster cell. Thus, we have altered the passage, as follows:*

*__Page 22, line 21 f.__: In particular, we can see that the relative values, i.e. the number of damage claims in relation to the number of buildings within the same raster cell, are not evenly distributed in space.*

- p.26, line 8-9: The same is true for Germany, but the evidence is lacking. Maybe this point should be made clearer - already in the introduction.

*We thank the referee for this interesting addition. As this statement holds true for at least Switzerland and Germany, we think it is safe to make a more general statement by omitting the explicit focus on Switzerland. Thus, we altered the passage as follows:*

*__Page 30, line 8 ff.__: There seems to be a consensus among practitioners and experts that SWFs are responsible for a large share of all flood damage. However, this perception does not stem from quantitative research, but rather from single case studies or practical experience and, thus, lacks evidence.*

- p.27, line 23: Why? Please mention your assessment criteria.

*We chose a balance between level of detail and smoothness based on visual comparison of the produced output maps. The finer the resolution, the patchier the picture gets, while coarser resolution spatially smooth the results. We specified this passage, as follows.*

*__Page 31, line 21 ff.__: For this study, we aggregated the data to regular grids and visually compared the corresponding maps. Fine resolutions produced patchy patterns, while local characteristics got lost with coarse resolutions. Thus, we chose a resolution of 3-by-3 km, which constitutes a balanced compromise between level of detail and smoothing.*

I am looking forward to the revised paper. Thank you for this interesting analysis.
* * *
**Final comments by the authors:**

Note that during the revision of the manuscript, we noted that the acknowledgement were not complete and not accurate enough. Therefore, we took the liberty to amend this section accordingly.

[revised manuscript text omitted]